# Stochastic Amortization: A Unified Approach to Accelerate Feature and Data Attribution

**Ian Covert**[*]
Stanford University
icovert@stanford.edu

**Chanwoo Kim**[*]
University of Washington
chanwkim@uw.edu

**Su-In Lee**[†]
University of Washington
suinlee@uw.edu

**James Zou**[†]
Stanford University
jamesz@stanford.edu

**Tatsunori Hashimoto**[†]
Stanford University
thashim@stanford.edu

## Abstract

Many tasks in explainable machine learning, such as data valuation and feature attribution, perform expensive computation for each data point and are intractable for large datasets. These methods require efficient approximations, and although amortizing the process by learning a network to directly predict the desired output is a promising solution, training such models with exact labels is often infeasible. We therefore explore training amortized models with noisy labels, and we find that this is inexpensive and surprisingly effective. Through theoretical analysis of the label noise and experiments with various models and datasets, we show that this approach tolerates high noise levels and significantly accelerates several feature attribution and data valuation methods, often yielding an order of magnitude speedup over existing approaches.

## 1 Introduction

Many tasks in explainable machine learning (XML) perform some form of costly computation for every data point in a dataset. For example, common tasks include assessing individual data points' impact on a model's accuracy [33], or quantifying each input feature's influence on individual model predictions [68]. Many of these techniques are prohibitively expensive: in particular, those with game-theoretic formulations have exponential complexity in the number of features or data points, making their exact calculation intractable [89, 4].

Accelerating these methods is therefore a topic of great practical importance. This has been addressed primarily with Monte Carlo approximations [96, 16, 74], which are faster than brute-force calculations but can be slow to converge and impractical for large datasets. Alternatively, a promising idea is to *amortize* the computation, or to approximate each data point's output with a learned model, typically a deep neural network [2]. For example, in the feature attribution context, we can train an *explainer model* to predict Shapley values that describe how each feature affects a classifier's prediction [50].

There are several reasons why amortization is appealing, particularly with neural networks: similar data points often have similar outputs, pretrained networks extract relevant features and can be efficiently fine-tuned, and if the combined training and inference time is low then amortization can be faster than computing the object of interest (e.g., data valuation scores) for the entire dataset. However, it is not obvious how to train such amortized models, because standard supervised learning requires a dataset of ground truth labels that can be intractable to generate. Our goal here is therefore to explore efficiently training amortized models when exact labels are costly. Our main insight is that

---

[*]Equal contribution. [†]Equal advising.

38th Conference on Neural Information Processing Systems (NeurIPS 2024).

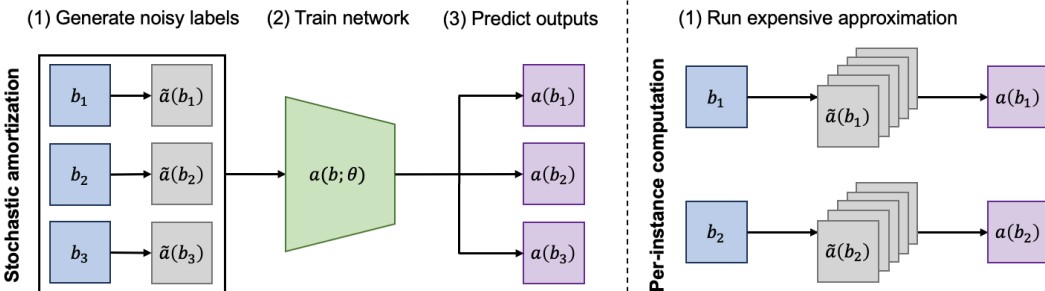

Figure 1: Diagram of stochastic amortization. Left: using a dataset with noisy labels $\tilde{a}(b)$ (e.g., images and data valuation estimates), we can train an amortized model that accurately estimates the true outputs $a(b)$ (e.g., data valuation scores). Right: the default approach of running an expensive approximation algorithm for each example (e.g., a Monte Carlo estimator with many samples [33]).

amortization is surprisingly effective with noisy labels: we train with inexpensive estimates of the true labels, and we find that this is theoretically justified when the estimates are unbiased.

We refer to this approach as *stochastic amortization* (Figure 1), and we find that it is applicable to a variety of XML tasks. In particular, we show that it is effective for feature attribution with Shapley values [68], Banzhaf values [11] and LIME [83]; for several formulations of data valuation [33, 35, 60, 103]; and to data attribution with datamodels [47]. Our experiments demonstrate significant speedups for several of these tasks: we find that amortizing across an entire dataset with noisy labels is often more efficient than current per-example approximations, especially for large datasets, and that amortized feature and data attribution models generalize well to unseen examples.

Our contributions in this work are the following:

- We present the idea of stochastic amortization, or training amortized models with noisy labels. We analyze the role of noise from the label generation process and show theoretically that it is sufficient to use unbiased estimates of the ground truth labels (Section 3). We find that non-zero bias in the labels leads to learning an incorrect function, but that variance in the labels plays a more benign role of slowing optimization.

- We identify a range of applications for stochastic amortization in XML. Our theory only requires unbiased estimation of the task's true labels, and we find that such estimators exist for several feature attribution and data valuation methods (Section 4).

- Experimentally, we test multiple estimators for Shapley value feature attributions and find that amortization works when the labels are unbiased (Section 5). We also verify that amortization is effective for Banzhaf values and LIME. For data valuation, we apply amortization to Data Shapley and show that it allows us to scale this approach to larger datasets than in previous works.

- Throughout our experiments, we also analyze the scaling behavior with respect to the amount of training data and the quality of the noisy labels. In general, we find that amortization is more efficient than per-example computation when the training set used for amortization contains at least a moderate number of data points (e.g., >1K for data valuation).

Overall, our work shows the potential for accelerating many computationally intensive XML tasks with the same simple approach: amortizing with noisy, unbiased labels.

## 2  Background

We first introduce the basic idea of amortization in ML, which we discuss in a general setting before considering any specific XML tasks. Consider a scenario where we repeat similar computation for a large number of data points. We represent this general setup with a context variable $b \in \mathcal{B}$ and a per-context output $a(b) \in \mathcal{A}$. For example, these can be an image and its data valuation score, or an image and its feature attributions. Amortization can be used with arbitrary domains, but we assume Euclidean spaces, or $\mathcal{A} \subseteq \mathbb{R}^m$ and $\mathcal{B} \subseteq \mathbb{R}^d$, because many XML tasks involve real-valued outputs.

The computation performed for each context $b \in \mathcal{B}$ can be arbitrary as well. In some cases $a(b)$ is an intractable expectation, in which case we would typically approximate it with a Monte Carlo estimator [96, 33]. In other situations it is the solution to an optimization problem [91, 2], in which case we can define $a(b)$ via a parametric objective $h : \mathcal{A} \times \mathcal{B} \mapsto \mathbb{R}$:

$$a(b) \equiv \underset{a' \in \mathcal{A}}{\arg \min} \ h(a'; b). \tag{1}$$

We do not require a specific formulation for $a(b)$ in this work, but we will see that both the expectation and optimization setups provide useful perspectives on our proposal of training with noisy labels.

In situations with repeated computation, our two options are generally to (i) perform the computation separately for each $b \in \mathcal{B}$, or (ii) amortize the computation by predicting the output with a learned model. We typically implement the latter with a neural network $a(b; \theta)$, and our goal is to train it such that $a(b; \theta) \approx a(b)$. In training these models, the main challenge occurs when the per-instance computation is costly: specifically, it is not obvious how to train $a(b; \theta)$ without a dataset of ground truth solutions $a(b)$, which can be too slow to generate for many XML methods [68, 33]. We address this challenge in Section 3, where we prove that amortization tolerates training with noisy labels.

## 2.1 Related work

The general idea of amortized computation captures many tasks arising in physics, engineering, control and ML [2]. For example, amortization is prominent in variational inference [56, 82], meta learning [38, 29, 80] and reinforcement learning [45, 64, 39]. In the XML context, many recent works have explored amortization to accelerate costly per-datapoint calculations [12, 108, 49, 50, 18, 19]. Some of these works are reviewed by [13], and we offer a detailed overview in Appendix A.

For feature attributions, two works propose predicting Shapley values by training with a custom weighted least squares loss [50, 18]; our simpler approach can use any unbiased estimator and resembles a standard regression task. Two other works suggest modeling feature attributions with supervised learning [87, 14]; these recommend training with exact or high-quality labels, whereas we recognize the potential to use noisy labels that can be generated orders of magnitude faster. Concurrently, Zhang et al. [110] proposed training with a custom estimator for Shapley value feature attributions; our work is similar but derives stochastic amortization algorithms for a range of settings, including the use of any unbiased estimator (Section 3) and usage for various XML tasks including data valuation (Section 4).

For data valuation, two works consider predicting data valuation scores with supervised learning [35, 36], but these also use near-exact labels that limit the applicability to large datasets. Concurrently, Li and Yu [65] propose a family of data valuation estimators and a learning-based approach analogous to [50] but for data valuation; our approach uses a simpler training loss that works with any unbiased estimator, and unlike [65] its memory usage does not scale with the dataset size. Separately, another line of work focuses on accelerating the model retraining step underlying most data valuation methods [57, 104, 107], and these are complementary to our approach. Finally, while there are works that accelerate data attribution with datamodels [78, 27], we are not aware of any that use amortization.[2]

More broadly, our proposal to train amortized models with noisy labels can be viewed as a version of stochastic optimization, or training with noisy gradients. This fundamental idea is widely used in machine learning [5], but to our knowledge we are the first to study the broad applicability of noisy labels for accelerating diverse XML tasks.

## 3 Stochastic Amortization

We now discuss how to efficiently train amortized models with noisy labels. Following Section 2, we present this as a general approach before focusing on a specific XML task. One natural idea is to treat amortization like a standard supervised learning problem: we can parameterize a model $a(b; \theta)$, adopt a distribution $p(b)$ over the context variable, and then train our model with the following objective,

$$\mathcal{L}_{\text{reg}}(\theta) = \mathbb{E} \left[ \|a(b; \theta) - a(b)\|^2 \right]. \tag{2}$$

---

[2]Datamodels [47] performs data attribution by fitting a linear regression model, but this is not amortization because the model cannot predict attributions for new data points (see Appendix B).

This approach is called *regression-based amortization* [2] because it reduces the problem to a simple regression task. The challenge is that this approach cannot be used when we lack a large dataset of exact labels $a(b)$, which is common for computationally intensive XML methods (Section 4).

A relaxation of this idea is to train the model with inexact labels (see Figure 1). We assume that these are generated by a noisy oracle $\tilde{a}(b)$, which is characterized by a distribution of outputs for each context $b \in \mathcal{B}$. For example, the noisy oracle could be a statistical estimator of a data valuation score [33]. With this, we can train the model using a modified version of Eq. (2), where we consider the loss in expectation over both $p(b)$ and the noisy labels $\tilde{a}(b)$:

$$\tilde{\mathcal{L}}_{\text{reg}}(\theta) = \mathbb{E}\left[\|a(b;\theta) - \tilde{a}(b)\|^2\right]. \tag{3}$$

It is not immediately obvious when this approach is worthwhile: if the noisy oracle is too inaccurate we will learn the wrong function, so it is important to choose the oracle carefully. We find that there are two properties of the oracle that matter, and these relate to its systematic error and noise level, or more intuitively its bias and variance. We denote these quantities as follows for a specific value $b$,

$$\text{B}(\tilde{a} \mid b) = \|a(b) - \mathbb{E}[\tilde{a}(b) \mid b]\|^2, \qquad \text{N}(\tilde{a} \mid b) = \mathbb{E}\left[\|\tilde{a}(b) - \mathbb{E}[\tilde{a}(b) \mid b]\|^2 \mid b\right],$$

and based on these we can also define the global measures $\text{B}(\tilde{a}) \equiv \mathbb{E}_p[\text{B}(\tilde{a} \mid b)]$ and $\text{N}(\tilde{a}) \equiv \mathbb{E}_p[\text{N}(\tilde{a} \mid b)]$ for the distribution over context variables $p(b)$.[3] These terms are useful because they reveal a relationship between the two amortization objectives. In general, the objectives are related by the following two-sided bound (see proof in Appendix D):

$$\left(\sqrt{\tilde{\mathcal{L}}_{\text{reg}}(\theta) - \text{N}(\tilde{a})} - \sqrt{\text{B}(\tilde{a})}\right)^2 \leq \mathcal{L}_{\text{reg}}(\theta) \leq \left(\sqrt{\tilde{\mathcal{L}}_{\text{reg}}(\theta) - \text{N}(\tilde{a})} + \sqrt{\text{B}(\tilde{a})}\right)^2. \tag{4}$$

This relationship shows that reducing $\tilde{\mathcal{L}}_{\text{reg}}(\theta)$ towards its minimum value $\text{N}(\tilde{a})$ is similar to training with $\mathcal{L}_{\text{reg}}(\theta)$, only with a disconnect introduced by the bias $\text{B}(\tilde{a})$. The bias represents a source of irreducible error, because in the limit $\tilde{\mathcal{L}}_{\text{reg}}(\theta) - \text{N}(\tilde{a}) \to 0$ we have $\mathcal{L}_{\text{reg}}(\theta) = \text{B}(\tilde{a})$. On the other hand, when $\text{B}(\tilde{a}) = 0$ we can see that $\mathcal{L}_{\text{reg}}(\theta) = \tilde{\mathcal{L}}_{\text{reg}}(\theta) - \text{N}(\tilde{a})$, which means that training with the noisy loss is equivalent and will recover the correct function asymptotically. This last equality is easy to see given an unbiased noisy oracle $\tilde{a}(b)$, but the more general relationship in Eq. (4) emphasizes how non-zero bias can be problematic and lead to learning an incorrect function.

Aside from the bias, the variance plays a role as well, not in determining the function we learn but in making the model's optimization unstable or require more noisy labels. To illustrate the role of variance, we present a theoretical result considering the simplest case of a linear model $a(b;\theta)$ trained with SGD, which shows that high label noise slows convergence (see proof in Appendix D).

**Theorem 1.** *Consider a noisy oracle $\tilde{a}(b)$ that satisfies $\mathbb{E}[\tilde{a}(b) \mid b] = \tilde{\theta}b$ with parameters $\tilde{\theta} \in \mathbb{R}^{m \times d}$ such that $\|\tilde{\theta}\|_F \leq D$. Given a distribution $p(b)$, define the norm-weighted distribution $q(b) \propto p(b) \cdot \|b\|^2$ and the terms $\Sigma_p \equiv \mathbb{E}_p[bb^\top]$ and $\Sigma_q \equiv \mathbb{E}_q[bb^\top]$. If we train a linear model $a(\theta;b) = \theta b$ with the noisy objective $\tilde{\mathcal{L}}_{\text{reg}}(\theta)$ using SGD with step size $\eta_t = \frac{2}{\alpha(t+1)}$, then the averaged iterate $\bar{\theta}_T = \sum_{t=1}^{T} \frac{2t}{T(T+1)}\theta_t$ at step $T$ satisfies*

$$\mathbb{E}[\tilde{\mathcal{L}}_{\text{reg}}(\bar{\theta}_T)] - \text{N}(\tilde{a}) \leq \frac{4\,\text{Tr}(\Sigma_p)\left(\text{N}_q(\tilde{a}) + 4\lambda_{\max}(\Sigma_q)D^2\right)}{\lambda_{\min}(\Sigma_p)(T+1)},$$

*where $\text{N}_q(\tilde{a}) \equiv \mathbb{E}_q[\text{N}(\tilde{a} \mid b)]$ is the noisy oracle's norm-weighted variance, and $\lambda_{\max}(\cdot)$, $\lambda_{\min}(\cdot)$ are the maximum and minimum eigenvalues.*

The bound in Theorem 1 shows that noise slows convergence by its presence in the numerator, and interestingly, it appears in the form of a weighted version $\text{N}_q(\tilde{a})$ that puts more weight on values with large norm $\|b\|$; this is a consequence of assuming a linear model, but we expect the general conclusion of label variance slowing convergence to hold even for neural networks. As a corollary, we can see that the rate in Theorem 1 applies directly to $\mathcal{L}_{\text{reg}}(\theta)$ when the noisy oracle is unbiased (see Appendix D). We note that very high noise levels can in principle prevent effective optimization, and this can be mitigated by either reducing the noisy oracle's variance or taking more steps $T$.

---

[3]$\text{N}(\tilde{a} \mid b)$ is equal to the trace of the conditional covariance $\text{Cov}(\tilde{a} \mid b)$, and $\text{N}(\tilde{a}) = \text{Tr}(\mathbb{E}[\text{Cov}(\tilde{a} \mid b)])$.

Overall, our analysis shows that amortization with noisy labels is possible, although perhaps more difficult to optimize than training with exact labels. We next show that unbiased estimates are available for many XML tasks (Section 4), and we later find that this form of amortization is consistently effective with the noise levels observed in practice (Section 5), even providing better accuracy than per-example estimation in compute-matched comparisons. As a shorthand, we refer to training with the noisy objective $\tilde{\mathcal{L}}_{\mathrm{reg}}(\theta)$ in Eq. (3) as *stochastic amortization*.

## 4 Applications to Explainable ML

We now consider XML tasks that can be accelerated with stochastic amortization. Rather than using generic variables $b \in \mathcal{B}$ and $a(b) \in \mathcal{A}$, this section uses an input variable $x \in \mathcal{X}$, a response variable $y \in \mathcal{Y}$, and a model $f$ or a measure of its performance. As we describe below, each application of stochastic amortization is instantiated by a noisy oracle that generates labels for the given task.

### 4.1 Shapley value feature attribution

One of the most common tasks in XML is feature attribution, which aims to quantify each feature's influence on an individual prediction. The Shapley value has gained popularity because of its origins in game theory [89, 68], and like many feature attribution methods is based on querying the model while removing different feature sets [17]. Given a model $f$ and input $x$ that consists of $d$ separate features $x = (x_1, \ldots, x_d)$, we assume that we can calculate the prediction $f(x_S) \in \mathbb{R}$ for any feature set $S \subseteq [d]$.[4] With this setup, the Shapley values $\phi_i(x) \in \mathbb{R}$ for each feature $i \in [d]$ are defined as:

$$\phi_i(x) = \frac{1}{d} \sum_{S \subseteq [d] \setminus \{i\}} \binom{d-1}{|S|}^{-1} \left( f(x_{S \cup \{i\}}) - f(x_S) \right). \tag{5}$$

These scores satisfy several desirable properties [89], but they are impractical to calculate due to the exponential summation over feature subsets. Our goal is therefore to learn an amortized model $\phi(x; \theta) \in \mathbb{R}^d$ to directly predict feature attribution scores, and for this we require a noisy oracle.

Many recent works have studied efficient Shapley value estimation [10], and we first consider noisy oracles derived from Eq. (5), which defines the attribution as the feature's expected marginal contribution. There are several unbiased statistical estimators that rely on sampling feature subsets or permutations [6, 96, 77, 74, 58], and following Section 3 we can use any of these for stochastic amortization. Our experiments use the classic permutation sampling estimator [96, 74], which approximates the values $\phi_i(x)$ as an expectation across feature orderings. We defer the precise definition of this noisy oracle to Appendix E, along with the other estimators used in our experiments.

Next, we also consider noisy oracles derived from an optimization perspective on the Shapley value. A famous result from Charnes et al. [7] shows that the Shapley values are the solution to the following problem (with abuse of notation we discard the solution's intercept term),

$$\phi(x) = \arg\min_{a \in \mathbb{R}^{d+1}} \sum_{S \subseteq [d]} \mu(S) \left( f(x_S) - a_0 - \sum_{i \in S} a_i \right)^2, \tag{6}$$

where we use a least squares weighting kernel defined as $\mu^{-1}(S) = \binom{d}{|S|}|S|(d - |S|)$. Several works have proposed approximating Shapley values by solving this problem with sampled subsets, either using projected gradient descent [92] or analytic solutions [68, 16]. Among these, we use KernelSHAP [68] and SGD-Shapley [92] as noisy oracles in our experiments. The first is an M-estimator whose bias shrinks as the number of sampled subsets grows [99], so we expect it to lead to effective amortization; the latter has been shown to have non-negligible bias [10], so our theory in Section 3 suggests that it should lead to learning an incorrect function when used for amortization.

### 4.2 Alternative feature attributions

Next, we consider two alternative feature attribution methods: Banzhaf values [4, 11] and LIME [83]. These are closely related to Shapley values and are similarly intractable [26, 40], but we find that they offer statistical estimators that can be used for stochastic amortization.

---

[4]There are many ways to do so [17], e.g., we can set features to their mean or use masked self-attention.

First, Banzhaf values assign the following scores to each feature for a prediction $f(x)$ [4]:

$$\phi_i(x) = \frac{1}{2^{d-1}} \sum_{S \subseteq [d]\setminus\{i\}} \left( f(x_{S\cup\{i\}}) - f(x_S) \right). \tag{7}$$

These differ from Shapley values only in their choice of weighting function, and they admit a range of similar statistical estimators. One option is the MSR estimator from [103], which is unbiased and re-uses all model evaluations for each feature attribution estimate. We adopt this as a noisy oracle in our experiments (see a precise definition in Appendix E), but several other options are available.

Second, LIME defines its attribution scores $\phi_i(x)$ as the solution to the following optimization problem, given a weighting kernel $\pi(S)$ and penalty term $\Omega$ [83]:[5]

$$\arg\min_{a\in\mathbb{R}^{d+1}} \sum_{S\subseteq[d]} \pi(S) \left( f(x_S) - a_0 - \sum_{i\in S} a_i \right)^2 + \Omega(a). \tag{8}$$

As our noisy oracle for LIME, we use the popular approach of solving the above problem for subsets sampled according to $\pi(S)$. Similar to KernelSHAP [68], this is an M-estimator whose bias shrinks to zero as the sample size grows [99], so we expect it to lead to successful amortization.

Aside from these methods, other costly feature interpretation methods rely on unbiased statistical estimators and can be amortized in a similar fashion [98, 62, 32]. We leave further investigation of these methods to future work.

## 4.3 Data valuation

Next, data valuation aims to quantify how much each training example affects a model's accuracy. We consider labeled examples $z = (x, y)$ and a training dataset $\mathcal{D} = \{z_i\}_{i=1}^n$, and we analyze each data point's value by fitting models to subsampled datasets $\mathcal{D}_T \subseteq \mathcal{D}$ with $T \subseteq [n]$ and calculating a measure of the model's performance $v(\mathcal{D}_T) \in \mathbb{R}$ (e.g., its 0-1 accuracy). This general approach was introduced by Ghorbani and Zou [33], who defined the Data Shapley scores $\psi(z_i) \in \mathbb{R}$ as follows:

$$\psi(z_i) = \frac{1}{n} \sum_{T\subseteq[n]\setminus\{i\}} \binom{n-1}{|T|}^{-1} (v(\mathcal{D}_T \cup \{z_i\}) - v(\mathcal{D}_T)). \tag{9}$$

Subsequent work generalized the approach in different ways, which we briefly summarize before considering amortization. For example, Wang and Jia [103] used the Banzhaf value rather than Shapley value, and Kwon and Zou [60] considered the case of arbitrary semivalues [75]. These correspond to adopting a different weighting over subsets in Eq. (9), and the general case can be written as follows for a normalized weighting function $w(k)$:[6]

$$\psi(z_i) = \sum_{T\subseteq[n]\setminus\{i\}} w(|T|) \left(v(\mathcal{D}_T \cup \{z_i\}) - v(\mathcal{D}_T)\right). \tag{10}$$

Next, another extension is the case of *distributional data valuation*. Ghorbani et al. [35] incorporate an expectation over the original dataset $\mathcal{D}$, which they show leads to well defined scores even for data points $z = (x, y)$ outside the training set. Given a distribution over datasets of size $|\mathcal{D}| = n - 1$ and a weighting function $w(k)$, this version defines the score $\psi(z) \in \mathbb{R}$ for arbitrary $z$ as follows:

$$\psi(z) = \mathbb{E}_{\mathcal{D}} \sum_{T\subseteq[n-1]} w(|T|) \left(v(\mathcal{D}_T \cup \{z\}) - v(\mathcal{D}_T)\right). \tag{11}$$

When using any of these methods in practice, the scores are difficult to calculate due to the intractable expectation across datasets. However, a crucial property they share is that they can all be estimated in an unbiased fashion, and these estimates can therefore be used as noisy labels for stochastic amortization. We focus on Data Shapley and Distributional Data Shapley in our experiments [33, 35], and we use the Monte Carlo estimator from Ghorbani and Zou [33] as a noisy oracle (see the precise

---

[5]Following [17], we only consider the version of LIME with a binary interpretable representation.

[6]For this to be a valid expectation, semivalues require that $\sum_{k=0}^{n-1} \binom{n-1}{k} w(k) = 1$. Note that Data Shapley adopts $w(k) = \binom{n-1}{k}^{-1}/n$ and Data Banzhaf adopts $w(k) = 1/2^{n-1}$.

definition in Appendix E). Doing so allows us to train an amortized valuation model $\psi(z; \theta) \in \mathbb{R}$ that accelerates valuation within our training dataset, and that can also be applied to external data, e.g., when selecting informative new data points for active learning [36].

Finally, Appendix B discusses amortization for the datamodels data attribution technique [47]. This method measures how much each training data point $z_i \in \mathcal{D}$ affects the prediction for an inference example $x \in \mathcal{X}$, and we show that the scores are equivalent to a simple expectation that can be estimated in an unbiased fashion. The datamodels scores can therefore be amortized by adopting these estimates as noisy labels, but we leave further investigation of this approach to future work.

## 5  Experiments

Our experiments apply stochastic amortization to several of the tasks discussed in Section 4. We consider both feature attribution and data valuation, for image and tabular datasets, and using multiple architectures for our amortized models, including fully-connected networks (FCNs), ResNets [43] and ViTs [24]. Full details are provided in Appendix F, including our exact models and hyperparameters.

Our goal in each experiment is to perform feature attribution or data valuation for an entire dataset, and to compare the accuracy of stochastic amortization to running existing estimators on each point. We adopt a noisy oracle for each task (e.g., a Monte Carlo estimator of data valuation scores), we then fit an amortized network with one noisy label per training example, and our evaluation focuses on the accuracy of the amortized predictions relative to the ground truth. Our ground truth is obtained by running the noisy oracle to near-convergence for a large number of samples: for example, we run KernelSHAP [68] for feature attribution with 1M samples, and the TMC estimator [33] for data valuation with 10K samples. We test amortization when using different numbers of training examples, and for both training and unseen external data to evaluate the model's generalization. We find that amortization often denoises and strongly improves upon the noisy labels, leading to a significant accuracy improvement for the same computational budget.

### 5.1  Feature attribution

We first consider Shapley value feature attributions. This task offers a diverse set of noisy oracles, and we consider three options: KernelSHAP [68], permutation sampling [74] and SGD-Shapley [92]. Among these, our theory from Section 3 suggests that the first two will be effective for amortization, while the third may not because it is not an unbiased estimator. We follow the setup from [18] and implement our amortized network with a pretrained ViT-B architecture [24], and we use the ImageNette dataset [46] with $224 \times 224$ images partitioned into 196 patches of size $14 \times 14$.

As a first result, Figure 2 compares our training targets to the amortized model's predictions. The predicted attributions are significantly more accurate than the labels, even for the noisiest setting with just 512 KernelSHAP samples. Figure 3 (left) quantifies the improvement from amortization, and we observe similar results for both KernelSHAP and permutation sampling (see Appendix G): in both cases the error is significantly lower than that of the noisy labels, and it remains lower and improves as the labels become more accurate. To contextualize our

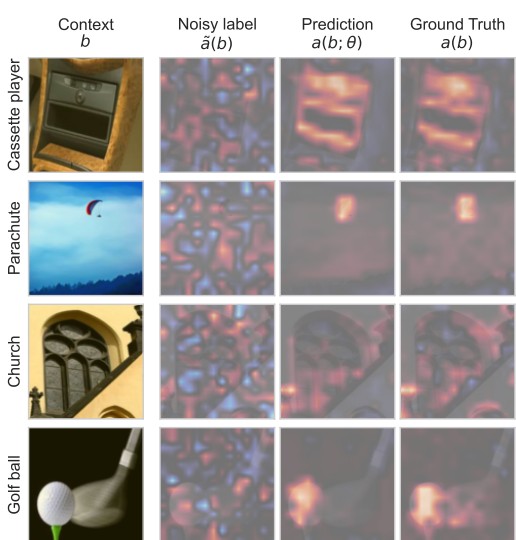

Figure 2: Stochastic amortization for Shapley value feature attributions. We compare the predicted attributions to the noisy labels and ground truth, which are generated using KernelSHAP with 512 and 1M samples, respectively.

amortized model's accuracy, we find that the error is similar to that of running KernelSHAP for 10-40K samples, even though our labels use an order of magnitude fewer samples (see Appendix G). We also find that the model generalizes to external data points (see Appendix G). In addition, we

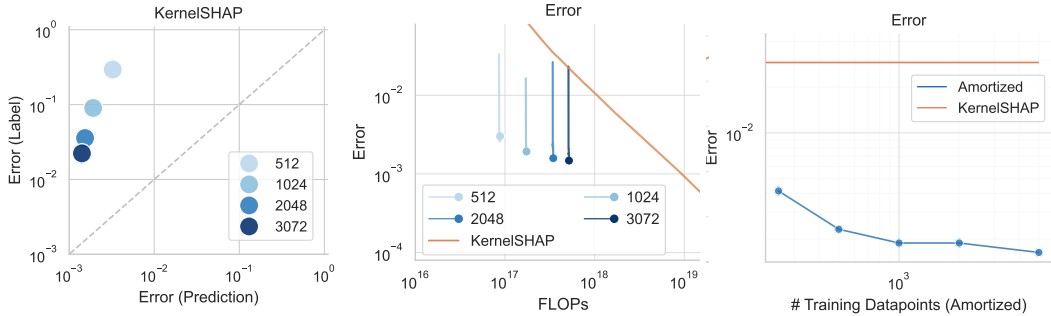

Figure 3: Amortized Shapley value feature attributions using KernelSHAP as a noisy oracle. Left: squared error relative to the ground truth attributions when using noisy labels with different numbers of samples (different noise levels). Center: estimation error as a function of FLOPs, where KernelSHAP incurs FLOPs via classifier predictions used to estimate the attributions, and amortization incurs additional FLOPs from training (training appears as a vertical line because the FLOPs are relatively low, and endpoints represent results from the final epoch). Right: estimation error with different training dataset sizes given equivalent compute per data point (matched by using fewer KernelSHAP samples when generating noisy labels for amortization and allowing up to 50 epochs of training).

show results for SGD-Shapley, where we confirm that it leads to poor amortization results due to its non-negligible bias (see Appendix G).

Next, we investigate the compute trade-off between calculating attributions separately (e.g., with KernelSHAP) and using amortization. Figure 3 (center-right) shows two results regarding this tradeoff. First, we measure the error as a function of FLOPs, where we account for the cost of generating labels and training the amortized model (see Appendix F). The FLOPs incurred by training are negligible compared to the KernelSHAP estimates, and we find that stopping at any time to fit an amortized model yields significantly better estimates. This suggests that amortization is an inexpensive final denoising step, regardless of how much compute was used for the noisy estimates.

Second, we test the effectiveness of amortization for different dataset sizes. We match the compute between the two scenarios, using 2440 KernelSHAP samples for per-example computation[7] and 2257 for amortization to account for the cost of training. This compute-matched comparison shows that amortization achieves lower estimation error for datasets ranging from 250-10K data points (Figure 3 right); it becomes more effective as the dataset grows, but it is useful even for small datasets.

Finally, Appendix G shows a comparison between stochastic amortization and FastSHAP [50, 18], an existing approach to amortized Shapley value estimation. We observe similar estimation accuracy in compute-matched comparisons, and find that both methods are significantly more accurate than per-example estimation (similar to Figure 3 center). Appendix G also show results for amortizing Banzhaf values and LIME: we find that amortization is more difficult for these methods due to the inconsistent scale of attributions between inputs, but we nonetheless observe an improvement in our amortized estimates versus the noisy labels.

## 5.2 Data valuation

Next, we consider data valuation with Data Shapley. For our noisy oracle, we obtain training labels by running the TMC estimator with different numbers of samples [33]. We first test our approach with the adult census and MiniBooNE particle physics datasets [25, 84], and following prior work we conduct experiments using versions of each dataset with different numbers of data points [53]. Our valuation model must predict scores for each example $z = (x, y)$, so we train FCNs that output scores for all classes and use only the relevant output for each data point.

As a first result, Figure 4 (left-center) shows the estimation accuracy for the MiniBooNE dataset when using 1K and 10K data points. The noisy estimates converge as we use more Monte Carlo samples, but we see that the amortized estimates are always more accurate in terms of both squared error and correlation with the ground truth. The improvement is largest for the noisiest estimates, where the amortized predictions have correlation $>0.9$ when using only 50 samples. Amortization

---

[7]This is the default number of samples for 196 features in the official repository: https://github.com/shap/shap.

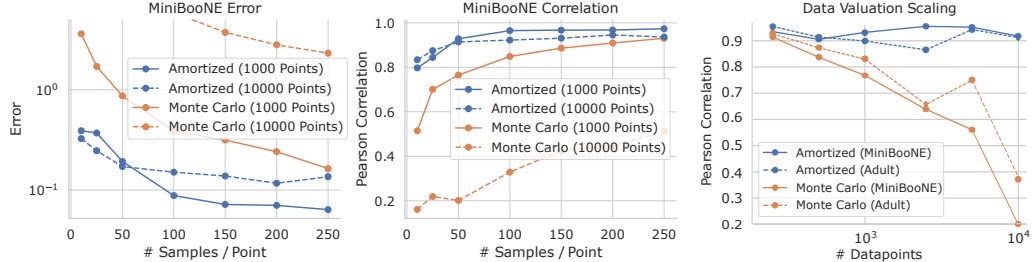

Figure 4: Amortized data valuation accuracy for tabular datasets. Left: mean squared error relative to the ground truth for the MiniBooNE dataset, normalized so that the mean valuation score has error equal to 1 (for 1K and 10K data points). The x-axis indicates how many Monte Carlo samples were used for each data point. Center: Pearson correlation with the ground truth for the MiniBooNE dataset (for 1K and 10K data points). Right: estimation accuracy for the MiniBooNE and adult census datasets as a function of dataset size (250 to 10K data points); we use 50 Monte Carlo samples per data point for all results and show the Pearson correlation with the ground truth.

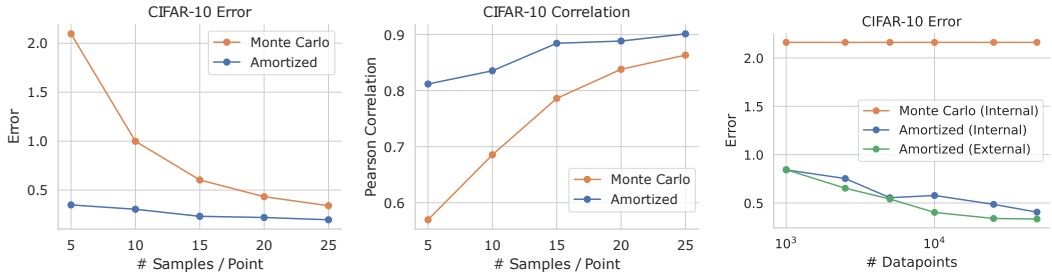

Figure 5: Distributional data valuation for CIFAR-10. Left: estimation error when using different numbers of samples for the noisy label estimates. Center: Pearson correlation with the ground truth for different numbers of noisy samples. Right: estimation error as a function of dataset size, where all results use 5 Monte Carlo samples per data points; we compare the error for amortized estimates on internal (training) and external (unseen) data points, demonstrating strong generalization.

is more beneficial for the 10K dataset, which suggests that training with more noisy labels can be a substitute for high-quality labels. Appendix G shows similar results with the adult census dataset.

Next, Figure 4 (right) considers the role of training dataset size for the estimates with 50 Monte Carlo samples. For both the adult census and MiniBooNE datasets, we see that the benefits of amortization are small with 250 data points but grow as we approach 10K data points. This number of samples is enough to maintain 0.9 correlation with the ground truth when using amortization, whereas the raw estimates become increasingly inaccurate. Stochastic amortization is therefore promising to scale data valuation beyond previous works, which typically focus on <1K data points [34, 60, 103].

### 5.3 Distributional data valuation

Finally, we consider Distributional Data Shapley [35], which is similar to the previous experiments but defines valuation scores even for points outside the training dataset; this allows us to test the generalization to unseen data. We use the CIFAR-10 dataset [59], which contains 50K training examples, and for our valuation model we train a ResNet-18 that outputs scores for each class.

Similar to the previous experiments, Figure 5 (left-center) evaluates the estimation accuracy for different numbers of Monte Carlo samples. We observe that the distributional scores converge faster because we use a smaller maximum dataset cardinality (see Appendix F), but that amortization still provides a significant benefit. The improvement is largest for the noisiest estimates, which in this case use just 5 samples: amortization achieves correlation 0.81 with the ground truth, versus 0.58 for the Monte Carlo estimates. The improvement is consistent across several measures of accuracy, including squared error, Pearson correlation and Spearman correlation (see Appendix G).

Next, we study generalization and the role of dataset size. We focus on the noisiest estimates with 5 Monte Carlo samples, because this low-sample regime is most relevant for larger datasets with millions of examples. We train the valuation model with different portions of the 50K training set, and we measure the estimation accuracy for both internal and unseen external data. Figure 5 (right) shows large improvements in squared error with as few as 1K data points, and we also observe improvement in correlation when we use at least 5K data points (10% of the dataset, see Appendix G). We additionally observe a small generalization gap, suggesting that the valuation model can be trained with a subset of the data and reliably applied to unseen examples.

Lastly, we test the usage of amortized valuation scores in downstream tasks. Following [53], the tasks we consider are identifying mislabeled examples and improving the model by removing low-value examples. These results are shown in Appendix G, and we find that the amortized estimates identify mislabeled examples more reliably than Monte Carlo estimates, and that filtering the dataset based on our estimates leads to improved performance.

## 6  Conclusion

This work explored the idea of stochastic amortization, or training amortized models with noisy labels. Our main finding is that fast, noisy supervision provides substantial compute and accuracy gains over existing XML approximations. This approach makes several feature attribution and data valuation methods more practical for large datasets and real-time applications, and it may have broader applications to amortization beyond XML [2]. Our proposal has certain limitations, including that stochastic amortization may become ineffective with sufficiently high noise levels, and that it is difficult to know a priori how much compute is necessary for label generation to achieve a desired error level in the amortized predictions.

Our work suggests multiple directions for future research. One direction is to study the trade-off between using a larger number of noisy labels or a smaller number of more accurate labels, which is a key difference from prior work that uses near-exact labels for amortization [14]. Other directions include scaling to datasets with millions of examples to test the limits of noisy supervision, leveraging more sophisticated data valuation estimators [106], using alternative model retraining primitives [57, 63, 37], and exploring amortization for other methods like datamodels (discussed in Appendix B).

## Code

We provide two repositories to reproduce each our results:

**Feature attribution**  https://github.com/chanwkimlab/amortized-attribution
**Data valuation**  https://github.com/iancovert/amortized-valuation

## Acknowledgements

The authors thank Yongchan Kwon for helpful discussions and advice on using OpenDataVal. We also thank Mukund Sudarshan and Neil Jethani for early conversations about amortizing data valuation. Chanwoo Kim and Su-In Lee were supported by the National Science Foundation (CAREER DBI-1552309 and DBI-1759487) and the National Institutes of Health (R35 GM 128638 and R01 AG061132).

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

# A  Extended Related Work

This section provides a more detailed review of amortization in XML. Many XML tasks involve analyzing individual data points, e.g., to determine the most important features or concepts in a model, or a data point's influence on the model's accuracy or an individual prediction. Among these methods, many require expensive per-example algorithms, and a trend in recent years has been to amortize this computation using deep learning. We discuss these works below, grouping them into several main XML problems that they address.

Within this discussion, we differentiate works not only based on their goal (e.g., feature attribution or data valuation) but also based on how they calculate the object of interest. For those that perform per-example computation, we distinguish between methods that solve a parametric optimization problem of the form $a(b) = \arg\min_{a'} h(a'; b)$ (e.g., with gradient descent) from those that exploit an analytic solution, either by directly calculating the result or using a Monte Carlo approximation. For those that use amortization, we distinguish those that use regression-based amortization with $\mathcal{L}_{\text{reg}}(\theta)$ from those that perform objective-based amortization with $\mathcal{L}_{\text{obj}}(\theta)$ (see Appendix C for more details on objective-based amortization [2]).

Table 1 summarizes the methods we discuss, including the various tasks they solve, the context and output domains $(\mathcal{A}, \mathcal{B})$ for each problem, and the computational approach. This perspective highlights the significant role of amortization in XML, and it also reveals opportunities for new applications, some of which we discuss in Section 4. Some of the same works are discussed by Chuang et al. [13], but we cover a wider range of methods, and we adopt the framing of Amos [2] by outlining the various parametric optimization problems and different amortization approaches.

**Feature attribution.** These methods aim to quantify the importance of a model's input features, typically for individual predictions (such methods are called *local* rather than *global* feature attributions, see Lundberg et al. [69]). The context is therefore a data example $x \in \mathcal{X}$, and the output is a vector of attributions in $\mathbb{R}^d$ when we have $d$ features. Feature attribution algorithms can be efficient to calculate, particularly when they are based on simple propagation rules [93, 97, 90, 95, 88, 111, 3] or a transformer's attention values [1]. However, another family of approaches are based on feature removal [17], and querying the model with many feature subsets is typically less efficient.

Among the feature removal-based methods, one of the most famous is Shapley value feature attributions [68, 89]. Two related methods are LIME [83] and Banzhaf values [11, 4], see our discussion of their similarity in Section 4. Many works have focused on efficiently calculating these attributions, because their computational complexity is exponential in the number of features. As discussed by Chen et al. [10], there are many stochastic estimators derived from an analytic expression for the attributions [6, 96, 77, 16, 74, 103, 58], and these are typically unbiased Monte Carlo estimators. Others are based on solving an optimization problem, for example with either M-estimation [83, 68, 105, 16] or gradient descent [92].

Lastly, there are also methods based on amortization, which offer the possibility of real-time feature attribution without a significant loss in accuracy. Two works consider objective-based amortization, which avoids the need for labels during training [50, 18]. Others consider regression-based amortization [86, 87, 14], and these recommend using high-quality labels; for example, Chuang et al. [14] use exact labels when possible. Our perspective is most similar to the concurrent work by Zhang et al. [110], in that we highlight the potential for regression-based amortization with inexpensive, noisy labels; but our work is more general in that we highlight the potential to use noisy labels from any unbiased estimator, and the applicability of this approach to other XML tasks like data valuation.

**Instance-wise feature selection.** A related set of methods select the most important features for individual predictions. The context in this case is an input $x \in \mathcal{X}$, and the output is a set of feature indices $S \in \mathcal{P}([d])$ where we use $\mathcal{P}(\cdot)$ to denote the power set. The optimal subset can be defined either based on the prediction $f(x_S)$ or $f(x_{[d] \setminus S})$ for a specific class [31], or based on the deviation from the original prediction $f(x)$ [12]. Among these methods, some solve the underlying optimization problem on a per-input basis [31, 30], and others solve it using objective-based amortization [21, 12, 108, 49]. These methods all require differentiating through discrete subsets, so they employ various continuous relaxations (e.g., the Concrete distribution [71]), as well as other tricks to constrain or penalize the subset size. These methods are sometimes thought of as attribution methods due to the continuous relaxations [31, 30], but we describe them as feature

Table 1: Summary of amortized methods in XML.

| Problem | Context | Domain | Analytic | Per-example optimization | $\mathcal{L}_{\mathrm{reg}}(\theta)$ | $\mathcal{L}_{\mathrm{obj}}(\theta)$ |
|---|---|---|---|---|---|---|
| Shapley value attribution | $x \in \mathcal{X}$ | $\mathbb{R}^d$ | Štrumbelj and Kononenko [96]
Okhrati and Lipani [77]
Mitchell et al. [74] | Lundberg and Lee [68]
Simon and Vincent [92]
Covert and Lee [16] | Schwarzenberg et al. [87]
Chuang et al. [14]
Zhang et al. [110] | Jethani et al. [50]
Covert et al. [18] |
| Leave-one-out attribution | | | Zeiler and Fergus [109] | | Schwab and Karlen [86] | |
| LIME attribution | | | | Ribeiro et al. [83] | | |
| Banzhaf value attribution | | | Covert et al. [17] | Chen and Jordan [11] | | |
| Instance-wise selection | $x \in \mathcal{X}$ | $\mathcal{P}([d])$ | | | | Chen et al. [12]
Yoon et al. [108]
Jethani et al. [49] |
| Image masking | | | | Fong and Vedaldi [31]
Fong et al. [30] | | Dabkowski and Gal [21] |
| Dynamic feature selection | $x_S \in \mathcal{X} \times \mathcal{P}([d])$ | $[d]$ | | Ma et al. [70] | He et al. [42] | Chattopadhyay et al. [9]
Covert et al. [19] |
| Counterfactual explanation | $x \in \mathcal{X}$ | $\mathcal{X}$ | | Wachter et al. [102] | | Mahajan et al. [72]
Verma et al. [101] |
| Data Shapley | $z_i \in \mathcal{D}$ | $\mathbb{R}$ | Ghorbani and Zou [33] | | Ghorbani et al. [35, 36] | Li and Yu [65] |
| Beta Shapley | | | Kwon and Zou [60] | | | |
| Data Banzhaf | | | Wang and Jia [103] | | | |
| Datamodels | $(z_i, x) \in \mathcal{D} \times \mathcal{X}$ | $\mathbb{R}$ | | | | |

selection methods because the final scores for each feature are restricted to the range $[0, 1]$ and are made continuous mainly for optimization purposes.

**Dynamic feature selection.** Next, other works select features separately for each prediction to achieve high accuracy with a small feature acquisition budget. Each selection is made based on a partially observed input, so the context is a feature subset $x_S$, which we write as belonging to the domain $\mathcal{X} \times \mathcal{P}([d])$, and the output is an index $i \in [d]$. There are several ways to define the optimal selection at each step, but one common approach is to define it as the feature with maximum conditional mutual information, or $i^* = \arg\max_i I(y; x_i \mid x_S)$, where the response variable $y$ and unobserved features $x_i$ are random variables and $x_S$ has a fixed observed value. Given access to this objective, it is trivial to solve by enumerating the possible indices; however, the mutual information is typically unavailable in practice and must be approximated. One approach is therefore to fit a proxy for the mutual information (e.g., via a generative model) and then optimize this proxy for each selection [70, 81, 8, 44]. An alternative is to learn a network that predicts the optimal selection at each step: He et al. [42] do so using imitation learning, which resembles regression-based amortization, and Chattopadhyay et al. [9] and Covert et al. [19] do so with an optimization-based view of the mutual information $I(y; x_i \mid x_S)$, which is objective-based amortization.

**Counterfactual explanation.** The goal of counterfactual explanations (also known as *recourse explanations*) is to identify small changes to an input that cause a desired change in a model's prediction. The context is an input $x \in \mathcal{X}$, and the output is a modified version $x' \in \mathcal{X}$ that is typically not too different from the original input. This family of methods was introduced by Wachter et al. [102] and is generally framed as an optimization problem involving the prediction $f(x')$ and a measure of the perturbation strength between $x'$ and $x$. Verma et al. [100] provide a review of these methods and the various choices for the optimization formulation. When computing these explanations, the most common approach is to solve the problem separately for each input, e.g., using gradient descent [102]. Other works have explored learning models that directly output the modified example $x'$, and these are typically implemented using objective-based amortization [72, 94, 101, 13].

**Data valuation.** The goal of data valuation is to assign scores to each training example that represent their contribution to a model's performance. The context is therefore a labeled training example $z \in \mathcal{D}$, and the output is a real-valued score. These methods are typically not defined via an optimization problem, but instead as a measure of the example's expected impact on performance across a distribution of preceding datasets [33, 35, 60, 103]. Existing methods therefore rely on analytic expressions for the valuation scores, which require Monte Carlo approximation because they are intractable. Two works have considered predicting data valuation scores given a dataset of near-exact estimates [35, 36], which is regression-based amortization. Our work is similar, but we use inexpensive estimates to reduce the cost of amortization and scale more efficiently to larger datasets. Concurrently, Li and Yu [65] proposed a learning-based approach analogous to FastSHAP for feature attribution [50], but whose memory requirements scale with the dataset size and limit applicability to large datasets. Besides these methods, there are also data valuation approaches that can be calculated without any approximations [51, 54, 61].

**Data attribution.** Finally, data attribution methods aim to quantify the effect of training examples on individual predictions. The context is therefore a training example $z \in \mathcal{D}$ paired with an inference example $x \in \mathcal{X}$, and the output is a real-valued attribution score. One classic approach to this problem is influence functions [15], which use a gradient-based approximation to avoid the cost of retraining [57, 37, 63]. Another class of methods involve measuring the effect of training with subsampled datasets [28, 47]. For datamodels, the main existing approximation algorithm is based on solving a global least squares problem [47], which does not correspond to any of the computational approaches listed in Table 1; notably, it is not a form of amortization because there is no model that can predict the attribution given a new tuple $(z, x)$. As an alternative to the global least squares problem, TRAK calculates similar scores to datamodels using a gradient-based approximation [78, 27]. We are not aware of any work investigating amortization for datamodels, but our results in Appendix B show how to approximate the scores based on their analytic solution, and how to amortize the computation by adopting Monte Carlo estimates as noisy training labels.

# B  Datamodels

The datamodels technique [47] aims to quantify how much each training data point $z_i \in \mathcal{D}$ affects the prediction for an inference example $x \in \mathcal{X}$. Similar to data valuation, the inference example's output given a training dataset $\mathcal{D}_T$ is represented by a function $v_x(\mathcal{D}_T)$, and the attribution scores $\zeta(z_i, x) \in \mathbb{R}$ are then defined as the solution to a joint least squares problem that can be solved after training many models with different datasets, as we show below.[8] For the inference example's output $v_x(\mathcal{D}_T)$, Ilyas et al. [47] focus on the loss rather than the raw prediction, but our perspective can accommodate any definition of this output.

Our insight on the datamodels technique is twofold. First, we show that the datamodels scores are equal to a simple expectation and can be estimated in an unbiased fashion. Second, we show that these calculations can be amortized by using the noisy Monte Carlo estimates as training targets. Our findings rely on a slight reformulation of the datamodels scores, where we deviate from Ilyas et al. [47] by using an intercept term and a subtly different dataset distribution: our distribution is biased towards size $nq$ for a value $q \in (0, 1)$ rather than using a fixed size, which was also considered in recent work by Saunshi et al. [85]. In the following result, we use the shorthand $\zeta(x) \equiv [\zeta(z_1, x), \dots, \zeta(z_n, x)]$ for the vector of data attributions. Given a probability $q \in (0, 1)$, we also define the weighting function $u(k) = q^k(1-q)^{n-1-k}$. With this setup, our reformulation is the following (see the proof in Appendix D).

**Proposition 1.** *Given a subset distribution that includes each data point $z_i \in \mathcal{D}$ with probability $q \in (0, 1)$, the data attribution scores defined by*

$$\zeta(x) \equiv \underset{a \in \mathbb{R}^{n+1}}{\arg\min} \ \mathbb{E}_T \left[ \left( a_0 + \sum_{i \in T} a_i - v_x(\mathcal{D}_T) \right)^2 \right]$$

*can be expressed as the following expectation:*

$$\zeta(z_i, x) = \sum_{T \subseteq [n] \setminus i} u(|T|) \left( v_x(\mathcal{D}_T \cup \{z_i\}) - v_x(\mathcal{D}_T) \right).$$

Perhaps surprisingly, this corresponds to a game-theoretic semivalue and reduces to the Banzhaf value when $q = 1/2$ [73]. Based on this perspective, we can estimate the score $\zeta(z_i, x)$ without solving the regression problem from Ilyas et al. [47]. For example, we can simply sample $k$ datasets $\mathcal{D}_T$ from the distribution with $q \in (0, 1)$, and then calculate the empirical average as follows:

$$\hat{\zeta}(z_i, x) = \frac{1}{k} \sum_{j=1}^{k} v_x(\mathcal{D}_{T_j} \cup \{z_i\}) - v_x(\mathcal{D}_{T_j} \setminus \{z_i\}).$$

Following Proposition 1, we have the unbiasedness property $\mathbb{E}[\hat{\zeta}(z_i, x)] = \zeta(z_i, x)$. Furthermore, rather than repeating this estimation for each attribution score, we can amortize the process by fitting an attribution model $\zeta(z_i, x; \theta)$ using the noisy estimates $\hat{\zeta}(z_i, x)$ as training targets. Our experiments do not test this approach, which we leave as a direction for future work. Implementing this requires a more complex architecture to handle two model inputs $x$ and $z$, but we speculate that if trained effectively, amortization could accelerate attribution within the training set and generalize to new inference examples $x \in \mathcal{X}$ with no additional overhead.

---

[8]As we noted in the main text, although datamodels performs data attribution by fitting a linear regression model, it is not a form of amortization because the resulting model cannot predict attribution scores $\zeta(z, x)$ for new datapoints. The attribution scores are given by the model's coefficients, not its predictions.

## C  Connection to Objective-Based Amortization

Recall that many tasks with repeated computation have an optimization view where $a(b) \equiv \arg\min_{a'} h(a'; b)$ [91, 2] (see Section 2). For completeness, this section describes an interpretation of stochastic amortization from this optimization perspective. An alternative to regression-based amortization when we have an objective $h(a'; b)$ that defines $a(b)$ is to train $a(b; \theta)$ directly with the $h$ objective:

$$\mathcal{L}_{\mathrm{obj}}(\theta) = \mathbb{E}[h(a(b; \theta); b)]. \tag{12}$$

This approach provides an alternative when exact labels are costly, but it can be unappealing because (i) the task may not offer a natural objective $h(a'; b)$, and (ii) minimizing $h(a'; b)$ may seem disconnected from the error $\|a(b; \theta) - a(b)\|$. We discuss these issues below and how they are related to stochastic amortization.

For issue (i), certain problems like Shapley values have a natural optimization characterization (see Section 4.1), but this is not always the case: for example, data valuation scores are framed as an expectation rather than via optimization (see Section 4.3 or [33]), so they lack an optimization view. We always have the trivial optimization perspective $h(a'; b) = \|a' - a(b)\|^2$, but this is not useful because it reduces $\mathcal{L}_{\mathrm{obj}}(\theta)$ to $\mathcal{L}_{\mathrm{reg}}(\theta)$ and requires the exact outputs $a(b)$. Regardless of whether the task offers a natural objective $h(a'; b)$, stochastic amortization can be viewed as defining a new objective that does not require the exact outputs: given a noisy oracle $\tilde{a}(b)$, stochastic amortization is equivalent to $\mathcal{L}_{\mathrm{obj}}(\theta)$ with the objective $h(a'; b) = \mathbb{E}\left[\|a' - \tilde{a}(b)\|^2\right]$, and if the oracle is unbiased then the optimal predictions are $a(b; \theta) = a(b)$.

Next, issue (ii) is a concern when $h(a'; b)$ is available but lacks a clear connection to the estimation error, which is directly reflected by $\mathcal{L}_{\mathrm{reg}}(\theta)$. The squared error is in many cases a more meaningful accuracy measure, and it is commonly used to evaluate estimation accuracy for feature attribution and data valuation [50, 103, 10]. Certain works on objective-based amortization have considered non-convex objectives $h(a'; b)$ where the exact output $a(b)$ is not well-defined [2], but we can understand the potential disconnect by focusing on a class of well behaved objectives. In particular, if we assume that $h(a'; b)$ is $\alpha$-strongly convex and $\beta$-smooth in $a'$ for all $b$, then the $\mathcal{L}_{\mathrm{reg}}(\theta)$ and $\mathcal{L}_{\mathrm{obj}}(\theta)$ objectives bound one other as follows,

$$\frac{\alpha}{2}\mathcal{L}_{\mathrm{reg}}(\theta) \leq \mathcal{L}_{\mathrm{obj}}(\theta) - \mathcal{L}_{\mathrm{obj}}^* \leq \frac{\beta}{2}\mathcal{L}_{\mathrm{reg}}(\theta), \tag{13}$$

where we define $\mathcal{L}_{\mathrm{obj}}^* \equiv \mathbb{E}[h(a(b); b)]$ (see proof in Appendix D.1). Eq. (13) shows that by minimizing $\mathcal{L}_{\mathrm{obj}}(\theta)$, we effectively minimize both upper and lower bounds on $\mathcal{L}_{\mathrm{reg}}(\theta)$. However, these bounds can be loose: for example, because we have $\mathcal{L}_{\mathrm{reg}}(\theta) \leq \frac{2}{\alpha}\mathcal{L}_{\mathrm{obj}}(\theta)$ and the strong convexity constant $\alpha$ shrinks to zero for Shapley values as the number of features grows [18], optimizing $\mathcal{L}_{\mathrm{obj}}(\theta)$ may become less effective in high dimensions. Stochastic amortization resolves this potential disconnect between $\mathcal{L}_{\mathrm{obj}}(\theta)$ and $\mathcal{L}_{\mathrm{reg}}(\theta)$ as follows: if we use an unbiased noisy oracle $\tilde{a}(b)$, then training with $\tilde{\mathcal{L}}_{\mathrm{reg}}(\theta)$ is equivalent to using $\mathcal{L}_{\mathrm{obj}}(\theta)$ with $h(a'; b) = \mathbb{E}\left[\|a' - \tilde{a}(b)\|^2\right]$, so we have $\alpha = \beta = 1$ in Eq. (13) and the regression objectives $\mathcal{L}_{\mathrm{reg}}(\theta)$ and $\tilde{\mathcal{L}}_{\mathrm{reg}}(\theta)$ are equal up to a constant $\mathcal{L}_{\mathrm{obj}}^* = \mathrm{N}(\tilde{a})$.

# D  Proofs

This section provides proofs for our claims in the main text. Appendix D.1 shows proofs for our results related to stochastic amortization in Section 3 and Appendix C, and then Appendix D.2 shows the proof for Proposition 1 related to datamodels.

## D.1  Amortization proofs

We first derive the inequality in Eq. (4) from the main text, which is the following:

$$\left(\sqrt{\tilde{\mathcal{L}}_{\text{reg}}(\theta) - \text{N}(\tilde{a})} - \sqrt{\text{B}(\tilde{a})}\right)^2 \leq \mathcal{L}_{\text{reg}}(\theta) \leq \left(\sqrt{\tilde{\mathcal{L}}_{\text{reg}}(\theta) - \text{N}(\tilde{a})} + \sqrt{\text{B}(\tilde{a})}\right)^2.$$

*Proof.* We begin by decomposing a per-input version of the noisy oracle loss $\tilde{\mathcal{L}}_{\text{reg}}(b; \theta)$, which takes an expectation over the noisy oracle $\tilde{a}(b)$ for a fixed $b$:

$$\begin{aligned}
\tilde{\mathcal{L}}_{\text{reg}}(b; \theta) &= \mathbb{E}\left[\|a(b; \theta) - \tilde{a}(b)\|^2 \mid b\right] \\
&= \|a(b; \theta) - \mathbb{E}[\tilde{a}(b) \mid b]\|^2 + \mathbb{E}[\|\tilde{a}(b) - \mathbb{E}[\tilde{a}(b) \mid b]\|^2 \mid b] \\
&= \|a(b; \theta) - \mathbb{E}[\tilde{a}(b) \mid b]\|^2 + \text{N}(\tilde{a} \mid b).
\end{aligned}$$

Next, we can also decompose a per-input version of the original regression loss $\mathcal{L}_{\text{reg}}(b; \theta)$ as follows using triangle inequality:

$$\begin{aligned}
\mathcal{L}_{\text{reg}}(b; \theta) &= \|a(b; \theta) - a(b)\|^2 \\
&\leq (\|a(b; \theta) - \mathbb{E}[\tilde{a}(b) \mid b]\| + \|a(b) - \mathbb{E}[\tilde{a}(b) \mid b]\|)^2 \\
&= \left(\sqrt{\tilde{\mathcal{L}}_{\text{reg}}(b; \theta) - \text{N}(\tilde{a} \mid b)} + \sqrt{\text{B}(\tilde{a} \mid b)}\right)^2.
\end{aligned}$$

Taking both sides of the inequality in expectation over $p(b)$, we arrive at the following upper bound:

$$\mathcal{L}_{\text{reg}}(\theta) \leq \mathbb{E}\left[\left(\sqrt{\tilde{\mathcal{L}}_{\text{reg}}(b; \theta) - \text{N}(\tilde{a} \mid b)} + \sqrt{\text{B}(\tilde{a} \mid b)}\right)^2\right].$$

One side of the bound in Eq. (4) comes from developing the square and applying Cauchy-Schwarz to the cross term:

$$\begin{aligned}
\mathcal{L}_{\text{reg}}(\theta) &\leq \mathbb{E}\left[\left(\sqrt{\tilde{\mathcal{L}}_{\text{reg}}(b; \theta) - \text{N}(\tilde{a} \mid b)} + \sqrt{\text{B}(\tilde{a} \mid b)}\right)^2\right] \\
&= \mathbb{E}\left[\tilde{\mathcal{L}}_{\text{reg}}(b; \theta) - \text{N}(\tilde{a} \mid b)\right] + \mathbb{E}\left[\text{B}(\tilde{a} \mid b)\right] + 2\mathbb{E}\left[\sqrt{\text{B}(\tilde{a} \mid b)\left(\tilde{\mathcal{L}}_{\text{reg}}(b; \theta) - \text{N}(\tilde{a} \mid b)\right)}\right] \\
&\leq \tilde{\mathcal{L}}_{\text{reg}}(\theta) - \text{N}(\tilde{a}) + \text{B}(\tilde{a}) + 2\sqrt{\text{B}(\tilde{a})\left(\tilde{\mathcal{L}}_{\text{reg}}(\theta) - \text{N}(\tilde{a})\right)} \\
&= \left(\sqrt{\tilde{\mathcal{L}}_{\text{reg}}(\theta) - \text{N}(\tilde{a})} + \sqrt{\text{B}(\tilde{a})}\right)^2.
\end{aligned}$$

This proves the upper bound in Eq. (4). For the lower bound, we return to the decomposition of the per-input noisy oracle loss $\tilde{\mathcal{L}}_{\text{reg}}(b; \theta)$ and apply triangle inequality as follows:

$$\tilde{\mathcal{L}}_{\text{reg}}(b; \theta) = \|a(b; \theta) - \mathbb{E}[\tilde{a}(b) \mid b]\|^2 + \text{N}(\tilde{a} \mid b)$$
$$\leq (\|a(b; \theta) - a(b)\| + \|a(b) - \mathbb{E}[\tilde{a}(b) \mid b]\|)^2 + \text{N}(\tilde{a} \mid b)$$
$$= \left( \sqrt{\mathcal{L}_{\text{reg}}(b; \theta)} + \sqrt{\text{B}(\tilde{a} \mid b)} \right)^2 + \text{N}(\tilde{a} \mid b).$$

Rearranging terms, we have

$$\mathcal{L}_{\text{reg}}(b; \theta) \geq \left( \sqrt{\tilde{\mathcal{L}}_{\text{reg}}(b; \theta) - \text{N}(\tilde{a} \mid b)} - \sqrt{\text{B}(\tilde{a} \mid b)} \right)^2,$$

and applying the same logic as above with Cauchy-Schwarz, we arrive at:

$$\mathcal{L}_{\text{reg}}(\theta) \geq \left( \sqrt{\tilde{\mathcal{L}}_{\text{reg}}(\theta) - \text{N}(\tilde{a})} - \sqrt{\text{B}(\tilde{a})} \right)^2.$$

Putting the two results together, this yields the two-sided bound in Eq. (4):

$$\left( \sqrt{\tilde{\mathcal{L}}_{\text{reg}}(\theta) - \text{N}(\tilde{a})} - \sqrt{\text{B}(\tilde{a})} \right)^2 \leq \mathcal{L}_{\text{reg}}(\theta) \leq \left( \sqrt{\tilde{\mathcal{L}}_{\text{reg}}(\theta) - \text{N}(\tilde{a})} + \sqrt{\text{B}(\tilde{a})} \right)^2.$$

$\square$

Next, we prove our SGD result for stochastic amortization in Theorem 1.

**Theorem 1.** *Consider a noisy oracle $\tilde{a}(b)$ that satisfies $\mathbb{E}[\tilde{a}(b) \mid b] = \tilde{\theta}b$ with parameters $\tilde{\theta} \in \mathbb{R}^{m \times d}$ such that $\|\tilde{\theta}\|_F \leq D$. Given a distribution $p(b)$, define the norm-weighted distribution $q(b) \propto p(b) \cdot \|b\|^2$ and the terms $\Sigma_p \equiv \mathbb{E}_p[bb^\top]$ and $\Sigma_q \equiv \mathbb{E}_q[bb^\top]$. If we train a linear model $a(\theta; b) = \theta b$ with the noisy objective $\tilde{\mathcal{L}}_{\text{reg}}(\theta)$ using SGD with step size $\eta_t = \frac{2}{\alpha(t+1)}$, then the averaged iterate $\bar{\theta}_T = \sum_{t=1}^{T} \frac{2t}{T(T+1)} \theta_t$ at step $T$ satisfies*

$$\mathbb{E}[\tilde{\mathcal{L}}_{\text{reg}}(\bar{\theta}_T)] - \text{N}(\tilde{a}) \leq \frac{4 \operatorname{Tr}(\Sigma_p) \left( \text{N}_q(\tilde{a}) + 4\lambda_{\max}(\Sigma_q) D^2 \right)}{\lambda_{\min}(\Sigma_p)(T+1)},$$

*where $\text{N}_q(\tilde{a}) \equiv \mathbb{E}_q[\text{N}(\tilde{a} \mid b)]$ is the noisy oracle's norm-weighted variance, and $\lambda_{\max}(\cdot)$, $\lambda_{\min}(\cdot)$ are the maximum and minimum eigenvalues.*

*Proof.* Given our assumption about the noisy oracle, we can re-write the noisy objective $\tilde{\mathcal{L}}_{\text{reg}}(\theta)$ as follows:

$$\tilde{\mathcal{L}}_{\text{reg}}(\theta) = \mathbb{E}\left[ \|(\theta - \tilde{\theta})b\|^2 \right] + \text{N}(\tilde{a}) = \operatorname{Tr}\left( (\theta - \tilde{\theta})^\top \Sigma_p (\theta - \tilde{\theta}) \right) + \text{N}(\tilde{a}).$$

It is clear that the minimizer is $\theta = \tilde{\theta}$ and that the minimum achievable error is $\tilde{\mathcal{L}}_{\text{reg}}(\tilde{\theta}) = \text{N}(\tilde{a})$. Because the objective composes a linear function with a convex function, it is convex in $\theta$ and we can analyze its convergence using a standard SGD result. Specifically, because we assume that $\|\tilde{\theta}\|_F \leq D$, we consider a projected SGD algorithm where each step is followed by a projection into the $D$-ball, or $\theta \leftarrow \theta \cdot \min(1, D/\|\theta\|_F)$. Note that it is common to assume a bounded solution space when proving SGD's convergence [76, 22, 5].

Before proceeding to the main convergence result, we must derive several properties of this objective and its stochastic gradients. The first is the strong convexity constant, and we begin by noting that the gradient is the following (see Section 2.5 of Petersen et al. [79] for a review of matrix derivatives):

$$\nabla \tilde{\mathcal{L}}_{\text{reg}}(\theta) = 2(\theta - \tilde{\theta})\mathbb{E}[bb^\top] = 2(\theta - \tilde{\theta})\Sigma_p.$$

For the strong convexity constant, we require a value $\alpha > 0$ such that the following is satisfied for all parameters $\theta, \theta'$:

$$\tilde{\mathcal{L}}_{\text{reg}}(\theta) \geq \tilde{\mathcal{L}}_{\text{reg}}(\theta') + \text{Tr}\left((\theta - \theta')^\top \nabla \tilde{\mathcal{L}}_{\text{reg}}(\theta')\right) + \frac{\alpha}{2}\|\theta - \theta'\|_F^2. \tag{14}$$

For the first two terms on the right side of the inequality, we can write:

$$\tilde{\mathcal{L}}_{\text{reg}}(\theta') + \text{Tr}\left((\theta - \theta')^\top \nabla \tilde{\mathcal{L}}_{\text{reg}}(\theta')\right) = \text{N}(\tilde{a}) + \text{Tr}\left((\theta' - \tilde{\theta})\Sigma_p(\theta' - \tilde{\theta})^\top\right) + 2\,\text{Tr}\left((\theta' - \tilde{\theta})\Sigma_p(\theta - \theta')^\top\right)$$
$$= \text{N}(\tilde{a}) + \text{Tr}\left((\theta - \tilde{\theta})\Sigma_p(\theta - \tilde{\theta})^\top\right) - \text{Tr}\left((\theta' - \theta)\Sigma_p(\theta' - \theta)^\top\right).$$

By using the minimum eigenvalue of $\Sigma_p$, we can write the following,

$$\text{Tr}\left((\theta' - \theta)\Sigma_p(\theta' - \theta)^\top\right) \geq \lambda_{\min}(\Sigma_p)\|\theta - \theta'\|_F^2,$$

and therefore satisfy Eq. (14) as follows:

$$\tilde{\mathcal{L}}_{\text{reg}}(\theta) = \text{N}(\tilde{a}) + \text{Tr}\left((\theta - \tilde{\theta})\Sigma_p(\theta - \tilde{\theta})^\top\right)$$
$$= \tilde{\mathcal{L}}_{\text{reg}}(\theta') + \text{Tr}\left((\theta - \theta')^\top \nabla \tilde{\mathcal{L}}_{\text{reg}}(\theta')\right) + \text{Tr}\left((\theta' - \theta)\Sigma_p(\theta' - \theta)^\top\right)$$
$$\geq \tilde{\mathcal{L}}_{\text{reg}}(\theta') + \text{Tr}\left((\theta - \theta')^\top \nabla \tilde{\mathcal{L}}_{\text{reg}}(\theta')\right) + \lambda_{\min}(\Sigma_p)\|\theta - \theta'\|_F^2.$$

We can therefore conclude that $\tilde{\mathcal{L}}_{\text{reg}}(\theta)$ is $2\lambda_{\min}(\Sigma_p)$-strongly convex.

The next property to derive is the expected gradient norm. When running SGD, we make updates using the following stochastic gradient estimate:

$$g(\theta) = 2\left(\theta b - \tilde{a}(b)\right)b^\top.$$

We require an upper bound on the stochastic gradient norm, which can be written as follows:

$$\mathbb{E}\left[\|g(\theta)\|_F^2\right] = \mathbb{E}\left[\|g(\theta) - \nabla \tilde{\mathcal{L}}_{\text{reg}}(\theta)\|_F^2\right] + \|\nabla \tilde{\mathcal{L}}_{\text{reg}}(\theta)\|_F^2. \tag{15}$$

For the first term in Eq. (15), which represents the gradient variance, we have:

$$\mathbb{E}\left[\|g(\theta) - \nabla \tilde{\mathcal{L}}_{\text{reg}}(\theta)\|_F^2\right] = 4\mathbb{E}\left[\|(\theta b - \tilde{a}(b))b^\top - (\theta - \tilde{\theta})\Sigma_p\|_F^2\right].$$

To find a simple expression for this term, we first consider the expectation over the distribution $\tilde{a}(b)$ with fixed $b \in \mathcal{B}$, which isolates label variation due to the noisy oracle:

$$\mathbb{E}_{\tilde{a}|b}\left[\|(\theta b - \tilde{a}(b))b^\top - (\theta - \tilde{\theta})\Sigma_p\|_F^2\right] = \mathbb{E}_{\tilde{a}|b}\left[\|(\tilde{a}(b) - \tilde{\theta}b)b^\top\|_F^2\right] + \|(\theta - \tilde{\theta})(bb^\top - \Sigma_p)\|_F^2$$
$$= \text{N}(\tilde{a} \mid b) \cdot \|b\|^2 + \|(\theta - \tilde{\theta})(bb^\top - \Sigma_p)\|_F^2.$$

When we take this in expectation over $p(b)$, the first term can be understood as a norm-weighted average of the noisy oracle's conditional variance:

$$\mathbb{E}\left[\mathrm{N}(\tilde{a} \mid b) \cdot \|b\|^2\right] = \int p(b)\mathrm{N}(\tilde{a} \mid b) \cdot \|b\|^2 db$$

$$= \mathrm{Tr}(\Sigma_p) \int \left(\frac{p(b) \cdot \|b\|^2}{\mathrm{Tr}(\Sigma_p)}\right) \mathrm{N}(\tilde{a} \mid b) db$$

$$= \mathrm{Tr}(\Sigma_p)\mathrm{N}_q(\tilde{a}).$$

The second term can be rewritten as follows using $\Sigma_p$ and $\Sigma_q$:

$$\mathbb{E}\left[\|(\theta - \tilde{\theta})(bb^\top - \Sigma_p)\|_F^2\right] = \mathbb{E}\left[\mathrm{Tr}\left((bb^\top - \Sigma_p)(\theta - \tilde{\theta})^\top(\theta - \tilde{\theta})(bb^\top - \Sigma_p)\right)\right]$$

$$= \mathbb{E}\left[\mathrm{Tr}\left((\theta - \tilde{\theta})(bb^\top - \Sigma_p)(bb^\top - \Sigma_p)(\theta - \tilde{\theta})^\top\right)\right]$$

$$= \mathrm{Tr}\left((\theta - \tilde{\theta})\mathbb{E}\left[(bb^\top - \Sigma_p)(bb^\top - \Sigma_p)\right](\theta - \tilde{\theta})^\top\right)$$

$$= \mathrm{Tr}\left((\theta - \tilde{\theta})(\mathrm{Tr}(\Sigma_p)\Sigma_q - \Sigma_p^2)(\theta - \tilde{\theta})^\top\right).$$

For the deterministic gradient upper bound in Eq. (15), we have:

$$\|\nabla \tilde{\mathcal{L}}_{\mathrm{reg}}(\theta)\|_F^2 = 4\,\mathrm{Tr}\left((\theta - \tilde{\theta})\Sigma_p^2(\theta - \tilde{\theta})^\top\right).$$

Putting these results together, we can upper bound the expected gradient norm for any parameter values $\|\theta\|_F \leq D$ as

$$\mathbb{E}\left[\|g(\theta)\|_F^2\right] = 4\,\mathrm{Tr}(\Sigma_p)\mathrm{N}_q(\tilde{a}) + 4\,\mathrm{Tr}(\Sigma_p)\,\mathrm{Tr}\left((\theta - \tilde{\theta})\Sigma_q(\theta - \tilde{\theta})^\top\right)$$

$$\leq 4\,\mathrm{Tr}(\Sigma_p)\mathrm{N}_q(\tilde{a}) + 4\,\mathrm{Tr}(\Sigma_p)\lambda_{\max}(\Sigma_q)\|\theta - \tilde{\theta}\|_F^2$$

$$\leq 4\,\mathrm{Tr}(\Sigma_p)\mathrm{N}_q(\tilde{a}) + 16\,\mathrm{Tr}(\Sigma_p)\lambda_{\max}(\Sigma_q)D^2.$$

Using our results for the objective's strong convexity and expected gradient norm, we can now invoke a standard SGD convergence result. Following Theorem 6.3 from Bubeck et al. [5], if we make $T$ updates with the specified step size, we have the following upper bound on the expected objective value:

$$\mathbb{E}[\tilde{\mathcal{L}}_{\mathrm{reg}}(\bar{\theta}_T)] - \mathrm{N}(\tilde{a}) \leq \frac{4\,\mathrm{Tr}(\Sigma_p)\mathrm{N}_q(\tilde{a}) + 16\,\mathrm{Tr}(\Sigma_p)\lambda_{\max}(\Sigma_q)D^2}{\lambda_{\min}(\Sigma_p)(T+1)}. \tag{16}$$

$\square$

Next, we prove a simple consequence of this result, which is that the rate from Theorem 1 applies to the original objective $\mathcal{L}_{\mathrm{reg}}(\theta)$ when we assume an unbiased noisy oracle.

**Corollary 1.** *Following the setup from Theorem 1, if the noisy oracle is unbiased, then the averaged iterate at step $T$ satisfies*

$$\mathbb{E}[\mathcal{L}_{\mathrm{reg}}(\bar{\theta}_T)] \leq \frac{4\,\mathrm{Tr}(\Sigma_p)\mathrm{N}_q(\tilde{a}) + 16\,\mathrm{Tr}(\Sigma_p)\lambda_{\max}(\Sigma_q)D^2}{\lambda_{\min}(\Sigma_p)(T+1)}.$$

*If the oracle is also noiseless, or if we assume access to the exact oracle $\tilde{a}(b) = a(b)$, then the averaged iterate at step $T$ satisfies*

$$\mathbb{E}[\mathcal{L}_{\mathrm{reg}}(\bar{\theta}_T)] \leq \frac{16\,\mathrm{Tr}(\Sigma_p)\lambda_{\max}(\Sigma_q)D^2}{\lambda_{\min}(\Sigma_p)(T+1)}.$$

*Proof.* The first result follows from combining Theorem 1 with the relationship between $\mathcal{L}_{\text{reg}}(\theta)$ and $\tilde{\mathcal{L}}_{\text{reg}}(\theta)$: if we assume the noisy oracle is unbiased, or $\text{B}(\tilde{a}) = 0$, then we have $\mathcal{L}_{\text{reg}}(\theta) = \tilde{\mathcal{L}}_{\text{reg}}(\theta) - \text{N}(\tilde{a})$, which implies the first inequality. The second inequality follows from setting $\text{N}_q(\tilde{a}) = 0$ in the upper bound. $\square$

Next, we provide a proof for Eq. (13) regarding the connection between regression- and objective-based amortization, which is the following:

$$\frac{\alpha}{2}\mathcal{L}_{\text{reg}}(\theta) \leq \mathcal{L}_{\text{obj}}(\theta) - \mathcal{L}_{\text{obj}}^* \leq \frac{\beta}{2}\mathcal{L}_{\text{reg}}(\theta).$$

*Proof.* This result relies on well known properties from convex optimization [41]. Consider a fixed context variable $b \in \mathcal{B}$. Strong convexity with $\alpha > 0$ means that for all $a, a' \in \mathcal{A}$ we have:

$$h(a;b) \geq h(a';b) + (a - a')^\top \nabla_a h(a';b) + \frac{\alpha}{2}\|a - a'\|^2.$$

Considering the optimal value $a' = a(b)$, this simplifies to:

$$\frac{\alpha}{2}\|a - a(b)\|^2 \leq h(a;b) - h(a(b);b). \tag{17}$$

Next, smoothness with $\beta > 0$ means that for all $a, a' \in \mathbb{R}^d$ we have:

$$\|\nabla_a h(a;b) - \nabla_a h(a';b)\|^2 \leq \beta\|a - a'\|.$$

Lemma 3.4 in Bubeck et al. [5] shows that $\beta$-smoothness also implies the following:

$$\left|h(a;b) - h(a';b) - (a - a')^\top \nabla_a h(a';b)\right| \leq \frac{\beta}{2}\|a - a'\|^2.$$

Considering the optimal value $a' = a(b)$, this simplifies to:

$$h(a;b) - h(a(b);b) \leq \frac{\beta}{2}\|a - a(b)\|^2. \tag{18}$$

The bounds in Eq. (13) follow from substituting $a$ for predictions from a model $a(b;\theta)$, and considering Eq. (17) and Eq. (18) in expectation across the distribution $p(b)$. $\square$

## D.2 Datamodels proofs

Before proving our main claim for datamodels in Proposition 1, we first prove a more general result. This version considers an arbitrary symmetric distribution $p(T)$ over datasets, rather than the specific distribution parameterized by $q \in (0, 1)$ used in Proposition 1. By a symmetric distribution, we mean one that satifies $p(T) = p(T')$ whenever $|T| = |T'|$.

As setup for our derivation, notice that when we assume a symmetric distribution $p(T)$, we can define the following probabilities that are identical for all indices $i, j \in [n]$:

$$
\begin{aligned}
p_1 &\equiv \Pr(i \in T) \quad \text{for } i \in [n] \\
p_2 &\equiv \Pr(i, j \in T) \quad \text{for } i \neq j \\
p_3 &\equiv \Pr(i \in T, j \notin T) \quad \text{for } i \neq j.
\end{aligned}
$$

Note that we have $p_1 = p_2 + p_3$. For example, given a uniform distribution $p(T)$ over subsets with size $|T| = k$, which is used by Ilyas et al. [47], we have:

$$p_1 = \frac{\binom{n-1}{k-1}}{\binom{n}{k}} = \frac{k}{n} \qquad p_2 = \frac{\binom{n-2}{k-1}}{\binom{n}{k}} = \frac{k(k-1)}{n(n-1)} \qquad p_3 = \frac{\binom{n-2}{k-1}}{\binom{n}{k}} = \frac{k(n-k)}{n(n-1)}.$$

Alternatively, if we have each data point included independently with probability $q \in (0,1)$, which is used by Saunshi et al. [85] and Proposition 1, then we have $p_1 = q$, $p_2 = q^2$, and $p_3 = q(1-q)$.

Our more general claim about datamodels with a symmetric distribution $p(T)$ is the following.

**Lemma 1.** *Given a symmetric distribution $p(T)$, the data attribution scores defined by*

$$\zeta(x) \equiv \arg\min_{a \in \mathbb{R}^{n+1}} \ \mathbb{E}_{p(T)}\left[\left(a_0 + \sum_{i \in T} a_i - v_x(\mathcal{D}_T)\right)^2\right]$$

*can be expressed as the expectation $\zeta(z_i, x) = \mathbb{E}[c_i(T)v_x(\mathcal{D}_T)]$, where we define the weighting function $c_i(T)$ as follows:*

$$c_i(T) \equiv \frac{1}{p_3}\left(\mathbb{1}(i \in T) - \frac{p_2 - p_1^2}{p_3 + n(p_2 - p_1^2)}|T| - \frac{p_1 p_3}{p_3 + n(p_2 - p_1^2)}\right).$$

*Proof.* We can write the problem's partial derivatives as follows, both for the intercept term $a_0$ and the coefficients $a_i$ for $i \in [n]$:

$$\frac{\partial}{\partial a_0}\mathbb{E}\left[\left(a_0 + a^\top \mathbf{1}_T - v_x(\mathcal{D}_T)\right)^2\right] = 2\left(a_0 + \mathbb{E}\left[\mathbf{1}_T^\top a\right] - \mathbb{E}\left[v_x(\mathcal{D}_T)\right]\right)$$
$$= 2\left(a_0 + p_1 \mathbf{1}_n^\top a - \mathbb{E}\left[v_x(\mathcal{D}_T)\right]\right)$$

$$\frac{\partial}{\partial a_i}\mathbb{E}\left[\left(a_0 + a^\top \mathbf{1}_T - v_x(\mathcal{D}_T)\right)^2\right] = 2\left(\mathbb{E}\left[\mathbb{1}(i \in T)(\mathbf{1}_T^\top a + a_0)\right] - \mathbb{E}\left[\mathbb{1}(i \in T)v_x(\mathcal{D}_T)\right]\right)$$
$$= 2\left(p_1 a_0 + p_2 \mathbf{1}_n^\top a + (p_1 - p_2)a_i - p_1 \mathbb{E}\left[v_x(\mathcal{D}_T) \mid i \in T\right]\right).$$

We use the shorthand notation $\bar{v} \in \mathbb{R}^{n+1}$ for a vector with entries $\bar{v}_0 = \mathbb{E}[v_x(\mathcal{D}_T)]$ and $\bar{v}_i = \mathbb{E}[v_x(\mathcal{D}_T) \mid i \in T]$ for $i \in [n]$. Combining this with the partial derivatives, we can derive an analytic solution $a^* \in \mathbb{R}^{n+1}$ by setting the derivative to zero,

$$\begin{pmatrix} 1 & p_1 \mathbf{1}_n^\top \\ p_1 \mathbf{1}_n & p_2 \mathbf{1}_n \mathbf{1}_n^\top + p_3 I_n \end{pmatrix} a^* - \begin{pmatrix} 1 & 0 \\ 0 & p_1 I_n \end{pmatrix} \bar{v} = 0,$$

which yields the following equation for the solution:

$$a^* = \begin{pmatrix} 1 & p_1 \mathbf{1}_n^\top \\ p_1 \mathbf{1}_n & p_2 \mathbf{1}_n \mathbf{1}_n^\top + p_3 I_n \end{pmatrix}^{-1} \begin{pmatrix} 1 & 0 \\ 0 & p_1 I_n \end{pmatrix} \bar{v}.$$

To find the required matrix inverse, we can combine the Sherman-Morrison formula with the formula for block matrix inversion. The formula for block matrix inversion is the following [79]:

$$\begin{pmatrix} \mathbf{A} & \mathbf{B} \\ \mathbf{C} & \mathbf{D} \end{pmatrix}^{-1} = \begin{pmatrix} \mathbf{A}^{-1} + \mathbf{A}^{-1}\mathbf{B}(\mathbf{D} - \mathbf{C}\mathbf{A}^{-1}\mathbf{B})^{-1}\mathbf{C}\mathbf{A}^{-1} & -\mathbf{A}^{-1}\mathbf{B}(\mathbf{D} - \mathbf{C}\mathbf{A}^{-1}\mathbf{B})^{-1} \\ -(\mathbf{D} - \mathbf{C}\mathbf{A}^{-1}\mathbf{B})^{-1}\mathbf{C}\mathbf{A}^{-1} & (\mathbf{D} - \mathbf{C}\mathbf{A}^{-1}\mathbf{B})^{-1} \end{pmatrix}.$$

We are mainly interested in the lower rows of this matrix because we do not use the learned intercept term. For the lower right matrix, we have the following:

$$(\mathbf{D} - \mathbf{C}\mathbf{A}^{-1}\mathbf{B})^{-1} = \left(p_2 \mathbf{1}_n \mathbf{1}_n^\top + p_3 I_n - p_1^2 \mathbf{1}_n \mathbf{1}_n^\top\right)^{-1}$$

$$= \left((p_2 - p_1^2)\mathbf{1}_n \mathbf{1}_n^\top + p_3 I_n\right)^{-1}$$

$$= \frac{1}{p_3} I_n - \frac{p_2 - p_1^2}{p_3(p_3 + n(p_2 - p_1^2))} \mathbf{1}_n \mathbf{1}_n^\top.$$

Next, for the lower left vector, we have the following:

$$-(\mathbf{D} - \mathbf{C}\mathbf{A}^{-1}\mathbf{B})^{-1}\mathbf{C}\mathbf{A}^{-1} = -\left(\frac{1}{p_3} I_n - \frac{p_2 - p_1^2}{p_3(p_3 + n(p_2 - p_1^2))} \mathbf{1}_n \mathbf{1}_n^\top\right) p_1 \mathbf{1}_n$$

$$= -\frac{p_1}{p_3}\mathbf{1}_n + \frac{np_1}{p_3} \frac{p_2 - p_1^2}{p_3 + n(p_2 - p_1^2)} \mathbf{1}_n$$

$$= -\frac{p_1}{p_3 + n(p_2 - p_1^2)} \mathbf{1}_n.$$

This yields the following solution for the optimal coefficients $a_i^*$ with $i \in [n]$:

$$a_i^* = \frac{p_1}{p_3}\bar{v}_i - \frac{p_1(p_2 - p_1^2)}{p_3(p_3 + n(p_2 - p_1^2))} \sum_{j \in [n]} \bar{v}_j - \frac{p_1}{p_3 + n(p_2 - p_1^2)}\bar{v}_0.$$

Based on this solution, we can design a function $c_i(T)$ so that we have the expectation $\mathbb{E}[c_i(T)v_x(\mathcal{D}_T)] = a_i^*$. We define the function as follows,

$$c_i(T) \equiv \frac{1}{p_3}\left(\mathbb{1}(i \in T) - \frac{p_2 - p_1^2}{p_3 + n(p_2 - p_1^2)}|T| - \frac{p_1 p_3}{p_3 + n(p_2 - p_1^2)}\right),$$

and it can be verified that this satisfies the required expectation.

$\square$

A similar derivation to Lemma 1 is possible when we omit an intercept term in the datamodels optimization problem, but we do not show the result here.

Finally, we prove the special case of Lemma 1 considered in Proposition 1.

**Proposition 1.** *Given a subset distribution that includes each data point $z_i \in \mathcal{D}$ with probability $q \in (0,1)$, the data attribution scores defined by*

$$\zeta(x) \equiv \arg\min_{a \in \mathbb{R}^{n+1}} \mathbb{E}_T\left[\left(a_0 + \sum_{i \in T} a_i - v_x(\mathcal{D}_T)\right)^2\right]$$

*can be expressed as the following expectation:*

$$\zeta(z_i, x) = \sum_{T \subseteq [n] \setminus i} u(|T|)\left(v_x(\mathcal{D}_T \cup \{z_i\}) - v_x(\mathcal{D}_T)\right).$$

*Proof.* Consider the weighting function $c_i(T)$ introduced in the proof for Lemma 1. In this case, we can use the fact that $p_2 = p_1^2$ and write the weighting function as follows:

$$c_i(T) = \frac{1}{p_3}\left(\mathbb{1}(i \in T) - p_1\right) = \begin{cases} \frac{1-p_1}{p_3} & i \in T \\ -\frac{p_1}{p_3} & i \notin T. \end{cases}$$

Using the fact that $p(T) = q^{|T|}(1-q)^{n-|T|}$ and the coefficient values $(1-p_1)/p_3 = q^{-1}$ and $-p_1/p_3 = -(1-q)^{-1}$, we arrive at the result in Proposition 1. $\square$

# E Noisy Oracles for XML Methods

This section describes the statistical estimators used for each XML task, which are briefly introduced in Section 4. Each estimator serves as a noisy oracle for the given task, which allows us to train amortized models with inexpensive supervision.

**Shapley values.** We use three statistical estimators for Shapley value feature attributions. The simplest is permutation sampling, where we average each feature's contribution across a set of sampled orderings [6, 96]. For this approach, we use $\rho$ to denote a permutation of the indices $[d]$, where we let $\rho(i) \subseteq [d] \setminus \{i\}$ denote the elements appearing before $i$ in the permutation. The Shapley value can be understood as the marginal contribution $f(x_{\rho(i) \cup \{i\}}) - f(x_{\rho(i)})$ averaged across all possible permutations [89], so our first estimator involves sampling $k$ permutations $\rho_1, \ldots, \rho_k$, and then calculating the following average for each feature:

$$\hat{\phi}_i(x) = \frac{1}{k} \sum_{j=1}^{k} f(x_{\rho_j(i) \cup \{i\}}) - f(x_{\rho_j(i)}). \tag{19}$$

Next, we consider two estimators based on the Shapley value's least squares view. Recall that the Shapley value is the solution to the following problem (where we discard the intercept term),

$$\phi(x) = \underset{a \in \mathbb{R}^{d+1}}{\arg\min} \sum_{S \subseteq [d]} \mu(S) \left( f(x_S) - a_0 - \sum_{i \in S} a_i \right)^2, \tag{20}$$

where we use a weighting kernel defined as $\mu^{-1}(S) = \binom{d}{|S|} |S| (d - |S|)$ [7]. The first estimator we use based on this perspective is KernelSHAP [68], which solves this problem using $k$ subsets $S_j \subseteq [d]$ sampled according to $\mu(S)$. The problem is convex but constrained due to the weighting terms $\mu([d]) = \mu(\varnothing) = \infty$, so it must be solved via the KKT conditions; we refer readers to Covert and Lee [16] for the closed-form solution. We also consider the SGD-Shapley approach from Simon and Vincent [92]: rather than solving Eq. (20) exactly given sampled subsets, this approach solves the problem iteratively with projected stochastic gradient descent. Unlike the other estimators, SGD-Shapley has been shown to have non-negligible bias [10]. We used a custom implementation for SGD-Shapley, and we used an open-source implementation of permutation sampling and KernelSHAP.[9]

**Banzhaf values.** When calculating Banzhaf value feature attributions, we adapt the *maximum sample re-use* (MSR) estimator from Wang and Jia [103], which was originally used for data valuation but is equally applicable to feature attribution. For a single example $x \in \mathcal{X}$, we generate predictions using $k$ subsets $S_j \subseteq [d]$ sampled uniformly at random. We then split the subsets into those that include or exclude each feature $i \in [d]$, and we estimate the Banzhaf value as follows:

$$\hat{\phi}_i(x) = \frac{1}{|\{j : i \in S_j\}|} \sum_{j : i \in S_j} f(x_{S_j}) - \frac{1}{|\{j : i \notin S_j\}|} \sum_{j : i \notin S_j} f(x_{S_j}). \tag{21}$$

The re-use of samples across all features makes this estimator more efficient than independent Monte Carlo estimates [103], and re-using samples in this way is simpler for Banzhaf values than for other methods like Shapley values [20, 58]. We used a custom implementation for this approach.

**LIME.** Similar to KernelSHAP, the most popular estimator for LIME feature attributions is based on approximately solving its weighted least squares problem. Following the problem formulation in Section 4.2, which we simplify by omitting the regularization term $\Omega$, we sample $k$ subsets $S_j \subseteq [d]$ uniformly at random and solve the following importance sampling version of the objective:

$$\hat{\phi}_1(x), \ldots, \hat{\phi}_d(x) = \underset{a \in \mathbb{R}^{d+1}}{\arg\min} \sum_{j=1}^{k} \pi(S_j) \left( a_0 + \sum_{i \in S_j} a_i - f(x_{S_j}) \right)^2. \tag{22}$$

---

[9] https://github.com/iancovert/shapley-regression

When doing so, we discard the intercept term, we ensure that $k$ is large enough to avoid singular matrix inversion, and we use the default weighting kernel $\pi(S)$ for images in the official LIME implementation.[10] We opt to sample subsets uniformly rather than according to $\pi(S)$ because this approach is used in the official implementation, and because the weighting kernel is similar to sampling subsets uniformly at random [66]. We used a custom implementation of this approach.

**Data valuation.** For the various data valuation methods discussed in Section 4.3, we use the simplest unbiased approximation, which is a Monte Carlo estimator that averages the performance difference across a set of sampled datasets. Given a labeled data point $z$, we sample $k$ datasets $\mathcal{D}_j \subseteq \mathcal{D}$ from the appropriate distribution and calculate the following empirical average:

$$\hat{\psi}(z) = \frac{1}{k} \sum_{j=1}^{k} v(\mathcal{D}_j \cup \{z\}) - v(\mathcal{D}_j). \tag{23}$$

This is similar to the TMC algorithm from Ghorbani and Zou [33], but we do not employ truncation; our approach can be used with truncation, but the noisy labels and valuation model predictions would both become biased. Similar to previous works, we also adopt a minimum subset cardinality to avoid training models with an insufficient number of data points. We implemented this approach using the OpenDataVal package [53]. More sophisticated estimators are available, like those that exploit stratification [106], re-use samples across all data points [103], or assume sparse attribution scores [52], but we leave exploration of these estimators to future work.

---

[10]https://github.com/marcotcr/lime

# F   Experiment Details

**Model architectures.** For the feature attribution experiments, we used the ViT-Base architecture [24] for the classifier, and we used a modified version of the architecture for the amortized attribution model: following the approach from Covert et al. [18], we added an extra self-attention layer and three fully-connected layers that operate on each token, so that the output contains an attribution score for each class. When estimating feature attributions, we make predictions with subsets of patches by setting the held-out patches to zero, and the classifier was fine-tuned with random masking to accommodate missing patches [17, 48, 50, 18].

For the data valuation experiments, we used a FCN with two hidden layers of size 128 for the tabular datasets, and we used a ResNet-18 architecture [43] for CIFAR-10. Valuation scores are defined for labeled examples $z = (x, y)$, so the valuation model must account for both the features $x$ and the class $y$ when making predictions; rather than passing $y$ as a model input, our architecture makes predictions simultaneously for all classes, and we only use the estimate for the relevant class. When generating noisy labels using the TMC estimator [33], we trained a logistic regression model on raw input features for the tabular datasets, and for CIFAR-10 we trained a logistic regression on pre-trained ResNet-50 features whose dimensionality was reduced with PCA.

**Hyperparameters.** For the feature attribution experiments, we optimized the models using the AdamW optimizer [67] with a linearly decaying learning rate schedule. The maximum learning rate was tuned using the validation loss, we trained for up to 100 epochs, and we selected the best model based on the validation loss. Due to the small scale of the noisy labels for Banzhaf and LIME feature attributions, we re-scaled the labels during training to improve the model's stability (see Appendix G.2), but this re-scaling was not necessary for Shapley values.

For the data valuation experiments, we optimized the models using Adam [55] with a cosine learning rate schedule. The maximum learning rate, the number of training epochs and the best model from the training run were determined using the validation loss. Due to the small scale of the noisy labels, we found it helpful to re-scale them during training to have standard deviation approximately equal to 1. The cost of performing multiple training runs is negligible compared to generating the noisy training labels, so it is not reflected in our plots comparing amortization to the Monte Carlo estimator (e.g., Figure 4).

**Pretraining.** We found that training the amortized models was faster and more stable when we initialized using pretrained architectures. For the feature attribution experiments, we initialized the model using the existing ViT-Base classifier weights. For data valuation, we initialized from a classifier trained using the entire dataset, where we used a FCN for the tabular datasets and a ResNet-18 for CIFAR-10. When adapting these models to their respective amortization tasks, the feature attribution models had freshly initialized output layers (one self-attention and three fully-connected), and the data valuation models had identical output layers that we re-initialized to zero.

**Validation loss.** For the feature attribution experiments, we calculated the validation loss using independent estimates generated with same estimator and the same number of samples used for the noisy training labels. For the data valuation experiments, we implemented the validation loss using independent Monte Carlo estimates of the data valuation scores. Our independent estimates use only 10 samples for the tabular datasets, and just one sample for CIFAR-10. Amortization provides a significant benefit over per-example calculations even after accounting for the cost of the validation loss.

**Noisy oracle details.** The estimators used for each task are described in detail in Appendix E. For the feature attribution experiments, the permutation sampling and KernelSHAP estimators only require the number of samples as hyperparameters, and we tested multiple values in our experiments (e.g., Figure 3). For SGD-Shapley, we tuned the algorithm in several ways to improve its performance: we used a constant rather than decaying learning rate, we took a uniform average over iterates, we calculated gradients using multiple subsets, and we used the paired sampling trick from Covert and Lee [16] to reduce the gradient variance. We also tuned the learning rate to a value that did not cause divergence for any examples (5e-4). As described in Appendix G.1, we observed that SGD-Shapley often had the lowest squared error among the three Shapley value estimators (Figure 11), but that it did not lead to successful amortization due to its non-negligible bias (Figure 10).

For the data valuation experiments, models were trained on each subsampled dataset by fitting a logistic regression model, either on raw input features for the tabular datasets or on pre-trained ResNet-50 features for CIFAR-10. For the Data Shapley experiments in Section 5.2 with tabular datasets, we sampled datasets without replacement, we used a minimum cardinality of 5 points and a maximum cardinality equal to the number of points $n$. For the Distributional Data Shapley experiments in Section 5.3 with CIFAR-10, we sampled datasets with replacement, and we used a minimum cardinality of 100 and maximum cardinality of 1000.

**Ground truth.** In all our experiments, the ground truth is obtained by running an existing per-instance estimator for a large number of samples, and we use enough samples to ensure that it has approximately converged to the exact result. For the Shapley value feature attribution experiments, we run KernelSHAP for 1M samples. For Banzhaf values, we run the MSR estimator for 1M samples. For LIME, we run the least squares estimator for 1M samples. For the data valuation experiments, we run the Monte Carlo estimator for 10K samples. These estimators are costly to run for a large number of samples, so we do so for only a small portion of the dataset: we calculate the ground truth for 100 examples for the feature attribution experiments. For the data valuation experiments, we use 250 examples for the tabular datasets and 500 for CIFAR-10.

**Metrics.** The performance metrics used throughout our experiments are related to the estimation accuracy, which we evaluate using our ground truth values. Other works have evaluated Shapley value feature attributions against other methods [50, 18] or used data valuation scores in a range of downstream tasks [53], but our focus is on efficient and accurate estimation. For the feature attribution experiments, we used the squared error distance, which we calculate across all attributions; we used Pearson and Spearman correlation, which are calculated individually for each flattened vector of attribution scores and then averaged across data points; and we used sign agreement, which is averaged across all attributions. For the data valuation experiments, we used squared error distance, which we normalized so the mean ground truth valuation score has error equal to 1 (i.e., we report the error divided by the variance in ground truth scores); and we used Pearson correlation, Spearman correlation and sign agreement, which were calculated using the vectors of estimated and ground truth valuation scores.

**Datasets.** We used publicly available, open-source datasets for our experiments. For the feature attribution experiments, we used the ImageNette dataset, a natural image dataset consisting of ten ImageNet classes [46, 23]. We partitioned the $224 \times 224$ inputs into 196 patches of size $14 \times 14$, and the dataset was split into a training set with 9469 examples, a validation set with 1962 examples, and a test set with 1963 examples. The validation set was used to perform early stopping, and the test set was only used to evaluate the model's performance on external data.

For the data valuation experiments, we used two tabular datasets from the UCI repository: the MiniBooNE particle classification dataset [84] and the adult census income classification dataset [25]. We obtained these using the OpenDataVal package [53], we used variable numbers of training examples ranging from 250 to 10K, and we reserved 100 examples for validation in each case; these examples are only used to evaluate the performance of models trained on subsampled datasets. We also used OpenDataVal to add label noise to $20\%$ of the training examples. For CIFAR-10, we used 50K examples for training and 1K for validation. We tested multiple levels of label noise for CIFAR-10 (Appendix G.3), and our main text results use $10\%$ label noise.

**Compute resources.** For the feature attribution experiments, we used a single GeForce RTX 2080Ti GPU to train the amortized models. Training an amortized model on ImageNette dataset for 50 epochs required roughly 4 hours. For the data valuation experiments, we used a single RTX A6000 to train amortized models, and these each trained in under an hour.

**FLOPs profiling.** When profiling compute for our feature attribution experiments (see Figure 3), we first measured the following constants using the DeepSpeed FLOPs profiler, all using the ViT-Base architecture:

- The classifier's forward pass requires 17,563,067,904 FLOPs $\approx$ 17.6 GFLOPs per prediction.

- The amortized model's forward pass requires 21,335,153,664 FLOPs $\approx$ 21.3 GFLOPs per prediction, due to the model having an extra transformer block for fine-tuning.

- The amortized model has 104,730,000 $\approx$ 105M trainable parameters.

Next, for the FLOPs comparison shown in Figure 3 we calculated total FLOPs for the two approaches as follows. For KernelSHAP, the total FLOPs = 17.6 GFLOPs × (number of subset samples) × (number of training datapoints). For the amortized models, the total FLOPs are the sum of multiple terms:

1. FLOPs for obtaining noisy labels = 17.6 GFLOPs × (number of subset samples) × (number of training datapoints)

2. FLOPs for the forward pass during training = 21.3 GFLOPs × (number of epochs) × (number of training datapoints)

3. FLOPs for the backward pass during training = 21.3 GFLOPs × 2 × (number of epochs) × (number of training datapoints), where 2 is a standard multiplier for calculating FLOPs during the backward pass

4. FLOPs for updating parameters = 105 MFLOPs × (number of epochs) × (number of training datapoints) / (batch size) × 19, where 19 is due to the optimizer state

In the calculation above, 1) dominates due to the large number of subset samples used for each datapoint (e.g., >512 for KernelSHAP). Meanwhile, the training epochs in 2), 3) and 4) are relatively low due to our models' fast convergence (<25 epochs). In the FLOPs calculation above, training an amortized model for one epoch is equivalent to obtaining predictions for roughly 3.65 additional subset samples per training datapoint. This is because $(3 \times 21{,}335{,}153{,}664 + 104{,}730{,}000 \times 19/64)/17{,}563{,}067{,}904 \approx 3.65$. In other words, the amount of compute for training an amortized model for up to 50 epochs translates to obtaining predictions for just 182.5 subset samples per datapoint, which is not enough to meaningfully improve the estimates by running KernelSHAP for more iterations. This explains why the lines for amortized models appear almost vertical in Figure 3 when we compared amortization to KernelSHAP. Our procedure for calculating FLOPs is similar to other works that profile computation when training neural networks.[11]

---

[11] We consulted the following resources: (i) https://www.lesswrong.com/posts/jJApGWG95495pYM7C/how-to-measure-flop-s-for-neural-networks-empirically, (ii) https://www.lesswrong.com/posts/fnjKpBoWJXcSDwhZk/what-s-the-backward-forward-flop-ratio-for-neural-networks, and (iii) https://www.adamcasson.com/posts/transformer-flops

# G  Additional Results

This section provides additional experimental results involving Shapley value feature attributions (Appendix G.1), Banzhaf value and LIME feature attributions (Appendix G.2), and data valuation (Appendix G.3).

## G.1  Shapley value amortization

First, Figure 6 shows that the performance of our amortized models is comparable to running KernelSHAP for 10-40k samples. For example, an amortized model trained on noisy KernelSHAP labels generated with 512 samples produces outputs of similar quality to running KernelSHAP for about 20k samples in terms of MSE, which is equivalent to a speedup of roughly 40x.

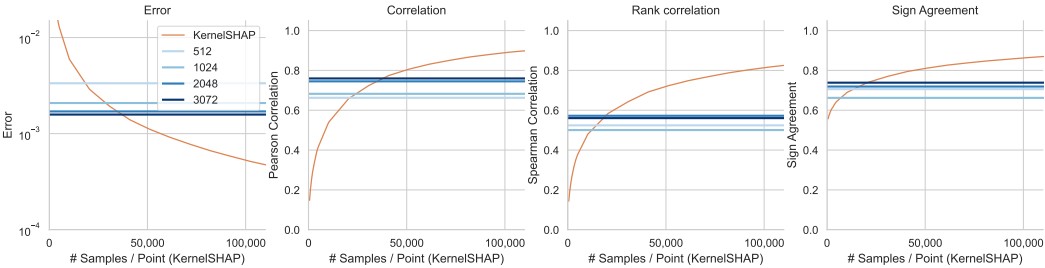

Figure 6: Comparison of the estimation accuracy between KernelSHAP and amortized predictions.

Next, Figure 7 shows an expanded version of Figure 3 (right) from the main text, where we compare amortization to per-instance estimation given a fixed amount of compute per data point. We observe that across four metrics, amortization provides a benefit over KernelSHAP even when training with a small portion of the ImageNette dataset.

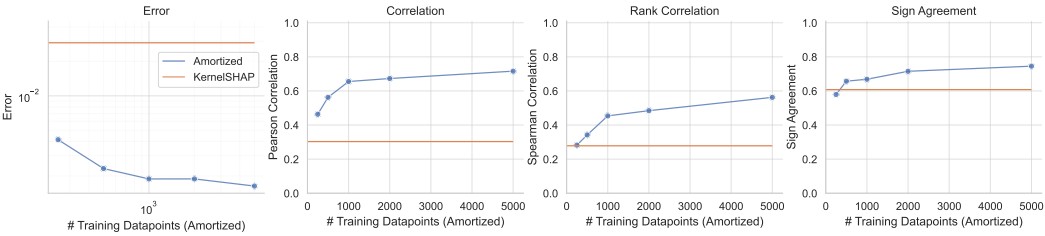

Figure 7: Estimation accuracy for amortization and KernelSHAP with different dataset sizes given equivalent compute.

Figure 8 shows a similar result, but the amortized model's performance is evaluated with external data points (i.e., points that are not seen during training). The benefits of amortization remain significant for most dataset sizes, reflecting that the model generalizes beyond the training data and can be used for real-time feature attribution with new examples.

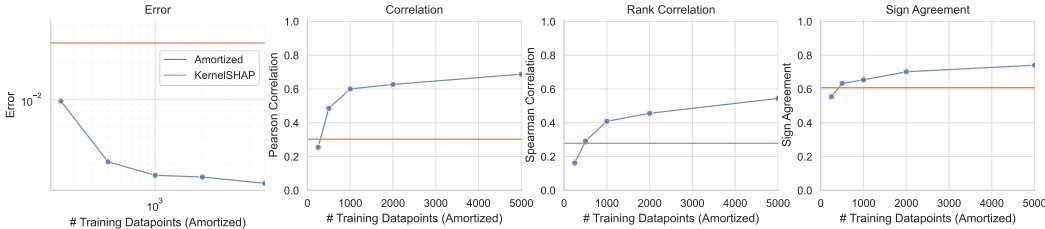

Figure 8: Estimation accuracy for amortization and KernelSHAP with different dataset sizes given equivalent compute (external data points).

Figure 9 shows an expanded version of Figure 3 (center) where we can see the benefits of amortization across multiple numbers of KernelSHAP samples when we account for the number of FLOPs.

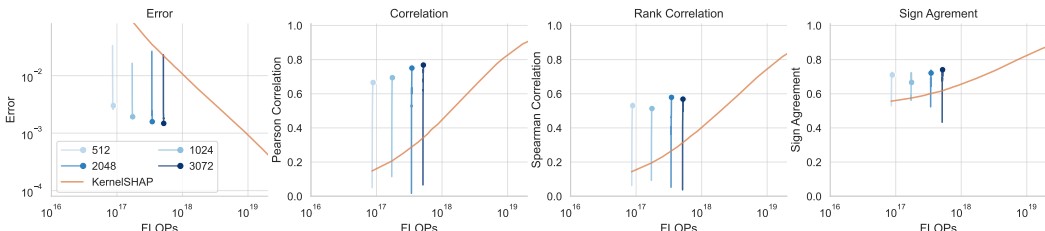

Figure 9: Error of amortization and KernelSHAP as a function of FLOPs.

Figure 10 shows an expanded version of Figure 3, where we can see the benefits of amortization for KernelSHAP and permutation sampling across four metrics. We also see that SGD-Shapley does not lead to successful amortization, because the predictions are worse than the noisy labels across all four metrics. This is perhaps surprising, because Figure 11 shows that SGD-Shapley has the lowest squared error among the three noisy oracles. The crucial issue with SGD-Shapley is that its estimates are not unbiased (see Section 3), an issue that has been shown in prior work [10].

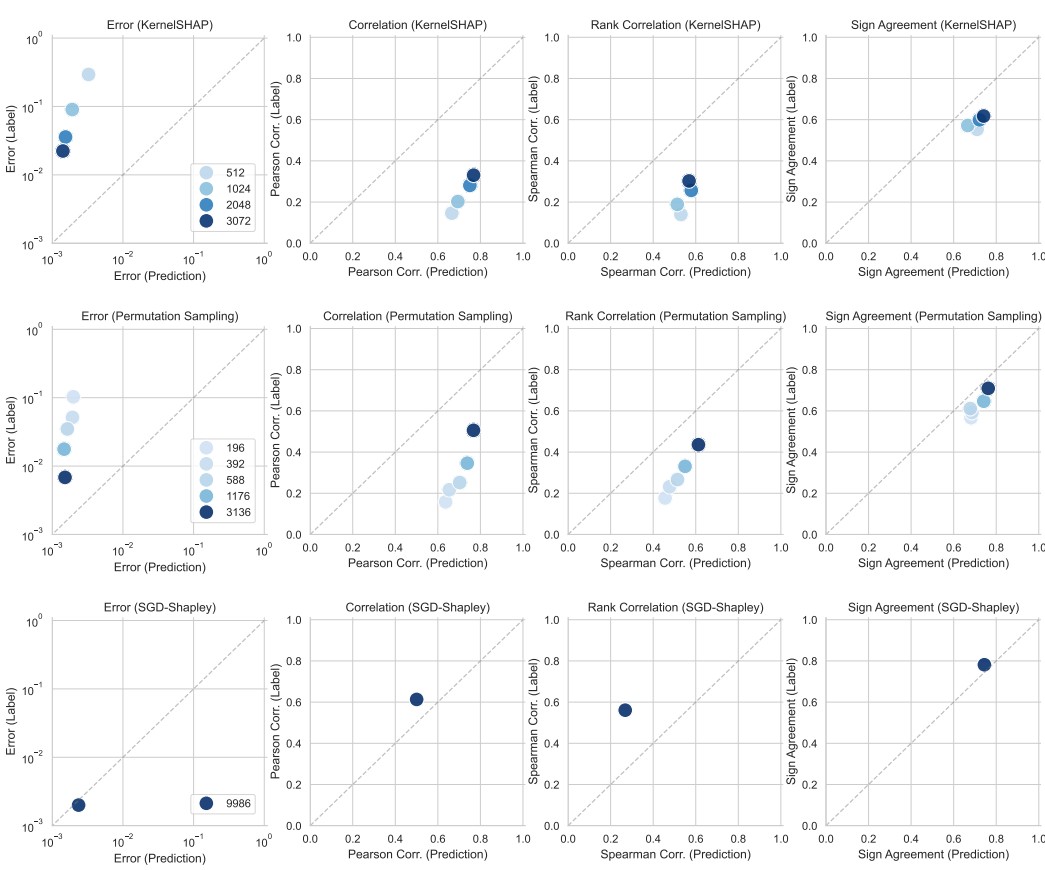

Figure 10: Comparison of the estimation error between noisy labels and amortized predictions for Shapley value feature attributions. Top: noisy labels generated using KernelSHAP with different numbers of samples. Middle: noisy labels generated using permutation sampling with different numbers of samples. Bottom: noisy labels generated using SGD-Shapley with different numbers of samples.

Figure 12 is similar to Figure 10, only the performance metrics are calculated using external points not seen during training. Like Figure 12, this result emphasizes that the amortized attribution model generalizes well and can be reliably used with new data points.

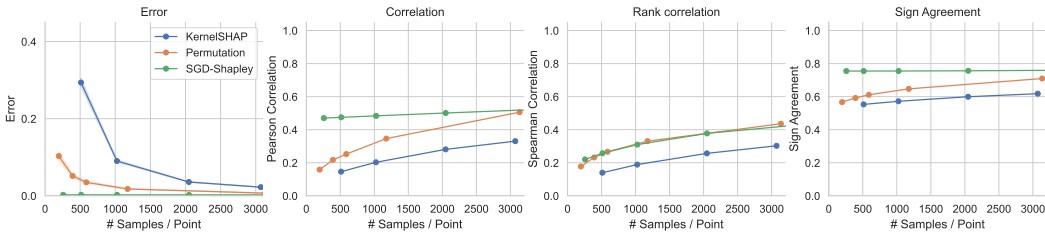

Figure 11: Comparison of the estimation error between different per-example estimators for Shapley value feature attributions across varying numbers of samples.

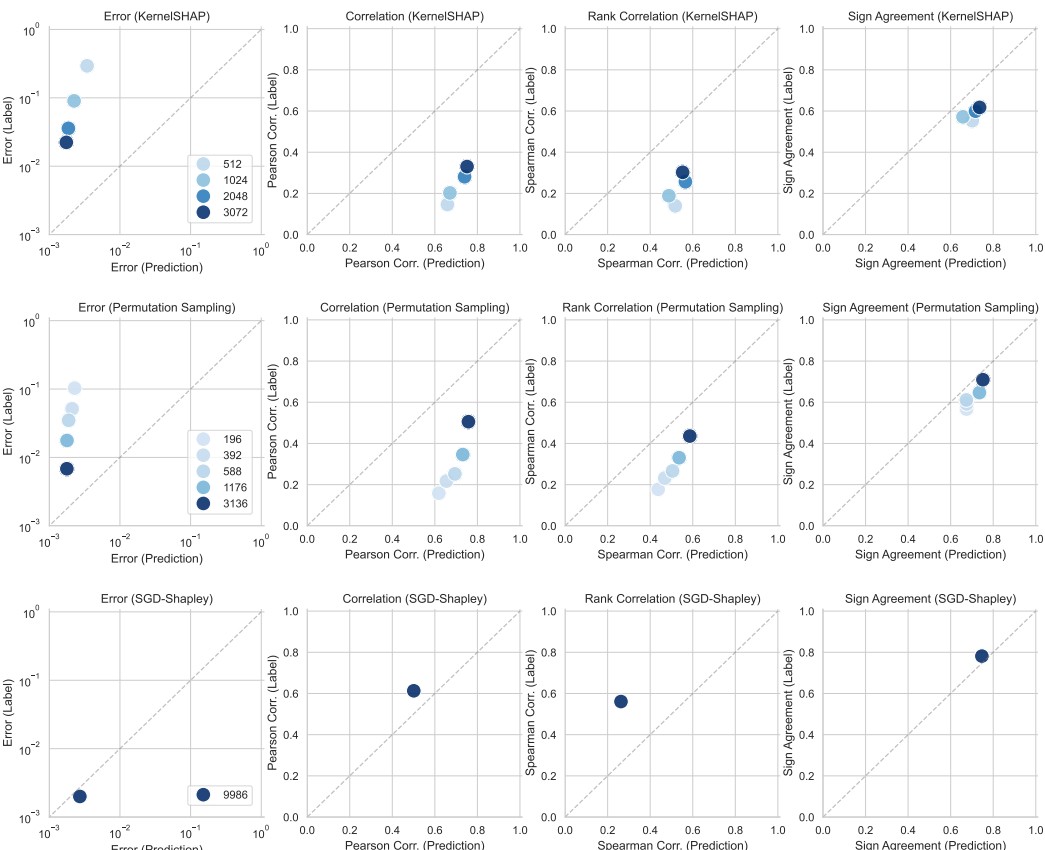

Figure 12: Comparison of the estimation error between noisy labels and amortized predictions for Shapley value feature attributions (external data points). Top: noisy labels generated using KernelSHAP with different numbers of samples. Middle: noisy labels generated using permutation sampling with different numbers of samples. Bottom: noisy labels generated using SGD-Shapley with different numbers of samples.

Finally, Figure 13 provides a comparison between stochastic amortization and FastSHAP [50, 18], one of the main existing approaches to amortized Shapley value estimation. For stochastic amortization, we use our previous results with permutation sampling as the noisy oracle and five different numbers of samples. For FastSHAP, we train the same ViT-Base architecture following the approach from Covert et al. [18], using 32 subset samples per gradient step. We monitor FastSHAP's estimation accuracy at the end of each epoch while training for a total of 100 epochs. We observe that FastSHAP and stochastic amortization achieve similar error at each total FLOPs budget, where both methods incur FLOPs from training the amortized model, stochastic amortization incurs FLOPs upfront when generating noisy labels, and FastSHAP incurs FLOPs during training while sampling subsets for each gradient step. FastSHAP is slightly more accurate at the largest computational budget, and both are significantly more accurate than per-sample estimation with KernelSHAP. Due to their similar performance, the main advantage of our approach is its simplicity (a standard regression objective rather than the custom FastSHAP objective), and the flexibility to use any unbiased estimator as the noisy oracle; future work may find that stochastic amortization is more effective with other noisy oracles that we did not try here.

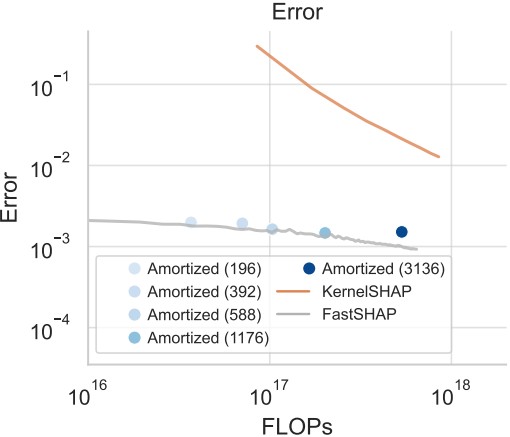

Figure 13: Comparison between stochastic amortization, FastSHAP and KernelSHAP as a function of total FLOPs.

## G.2   Banzhaf value and LIME amortization

As we discussed in Section 4.2, Banzhaf value and LIME feature attributions are two XML tasks closely related to Shapley values that can be amortized in a similar fashion. Both are intractable to calculate exactly, but they have (approximately) unbiased estimators that can be used as noisy labels for stochastic amortization. Appendix E describes these estimators, namely the MSR estimator for Banzhaf values [103] and the least squares estimator for LIME [83]. Prior work has also shown that these approaches generate similar outputs [66], although the standard computational approaches are quite different.

In amortizing these methods, one trend we observed is that Banzhaf value and LIME feature attributions have a very different scale from Shapley values. Figure 14 shows that they not only have smaller norm, but that the distribution is concentrated at small magnitudes with a long tail of larger magnitudes. This is troublesome, because our theory is focused on the squared error $\|a(b;\theta) - a(b)\|^2$ (see Section 3), which is dominated by the small number of examples with large magnitudes. As a result, an amortized attribution model can appear to train well by accurately estimating attributions with a large norm, and predicting the remaining attributions to be roughly zero; in doing so it may fail to provide the correct relative feature ordering for most examples, which is more important for practical usage of feature attributions.

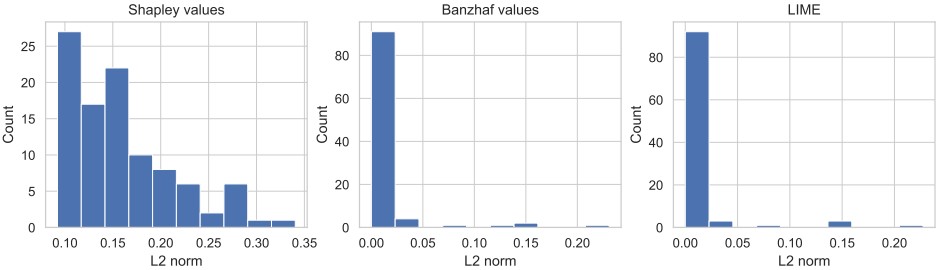

Figure 14: Comparison of the distribution of norms between different feature attribution methods. We plotted the L2 norm of the 100 ground truths for each feature attribution.

The issue described above is precisely what occurs when we amortize Banzhaf value feature attributions, as shown in the top row of Figure 15. We observe that the squared error from our predictions is lower than that of the noisy labels, which is consistent with our theory from Section 3 and our results for Shapley values (Appendix G.1). However, the amortized estimates perform worse than the noisy labels on the remaining metrics, particularly for the correlation scores that evaluate the relative feature ordering in each example's attributions (see details for the metrics in Appendix F).

To alleviate this issue, we experimented with a per-label normalization that eliminates the long tail of large attribution norms that dominate training: we simply normalized each example's attributions for each class to have a norm of 1. The results are shown in the bottom row of Figure 15, where we see that the predictions have higher correlation scores than the noisy labels. The squared error is higher for the amortized predictions, and the sign agreement is roughly the same. One issue with this heuristic is that the normalized noisy labels are biased: the normalized noisy label is not an unbiased estimator of the normalized exact label, because the normalization constant is not known a priori and must be calculated using the noisy label. However, the results show that amortization can to some extent denoise inexact labels even with this non-zero bias.

Figure 16 shows similar results as Figure 15 but with LIME feature attributions. We observe the same improvement in squared error from amortization, and a similar degradation in the correlation metrics (Figure 16 top). We then apply the same per-label normalization trick, and we observe a similar modest improvement in the correlation metrics, at least for noisier settings of the least squares estimator (Figure 16 bottom).

Overall, these results show that for the purpose of amortization, the consistency in scale of Shapley values is an unexpected advantage over Banzhaf values and LIME. This property is due in part to the Shapley value's efficiency axiom [89], which guarantees that the Shapley values sum to the difference between the prediction with all features and no features [68]. In comparison, LIME and Banzhaf values focus on marginal contributions involving roughly half of the features, which in many cases are near zero when the prediction is saturated; for example, previous works have observed that for

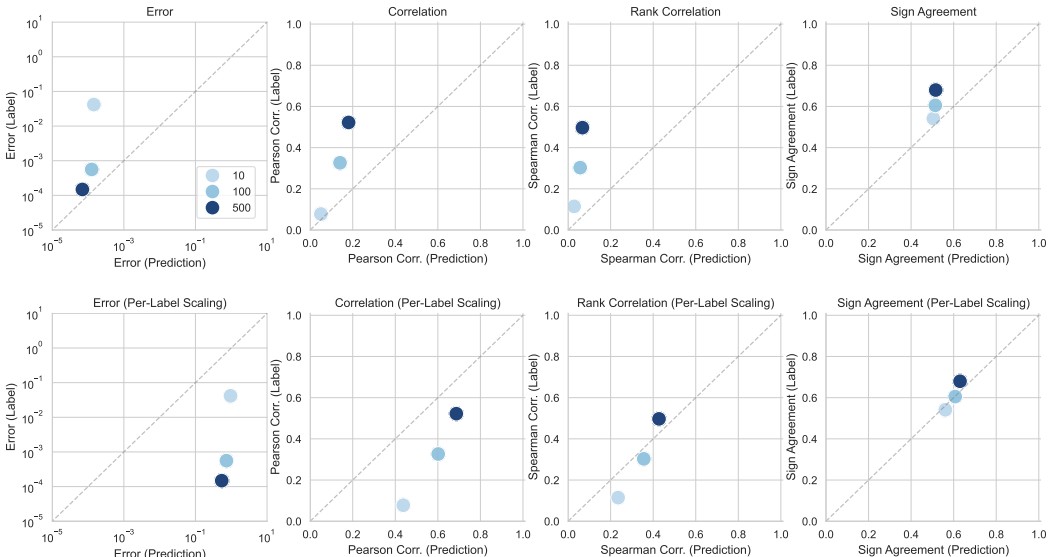

Figure 15: Comparison of the estimation error between noisy labels and amortized predictions for Banzhaf value feature attributions. Noisy labels were generated using the MSR estimator with different numbers of samples. Top: using raw estimates as noisy training labels. Bottom: normalizing each noisy training label separately to have unit norm for each class.

natural images containing relatively large objects, a large portion of the patches must be removed before we observe significant changes in the prediction [48, 18].

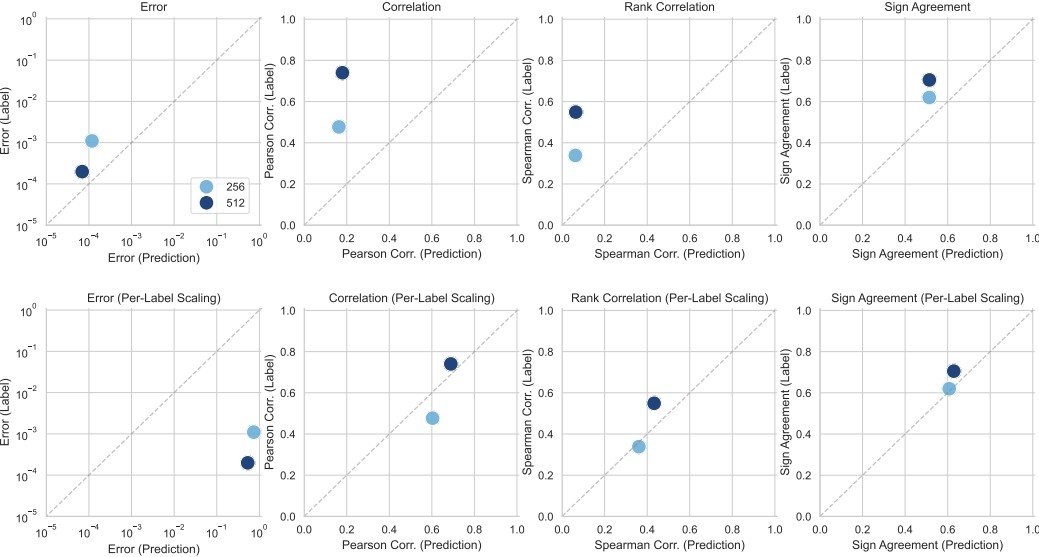

Figure 16: Comparison of the estimation error between noisy labels and amortized predictions for LIME feature attributions. Noisy labels were generated using LIME's least squares estimator with different numbers of samples. Top: using raw estimates as noisy training labels. Bottom: normalizing each noisy training label separately to have unit norm for each class.

### G.3 Data valuation

For the data valuation experiments, our first additional result compares amortization to the Monte Carlo estimator when using the MiniBooNE and adult datasets with different numbers of data points. The results are shown in Figure 17 and Figure 18, where we use datasets ranging from 1K to 10K points, and we see that the benefits of amortization increase with the size of the dataset.

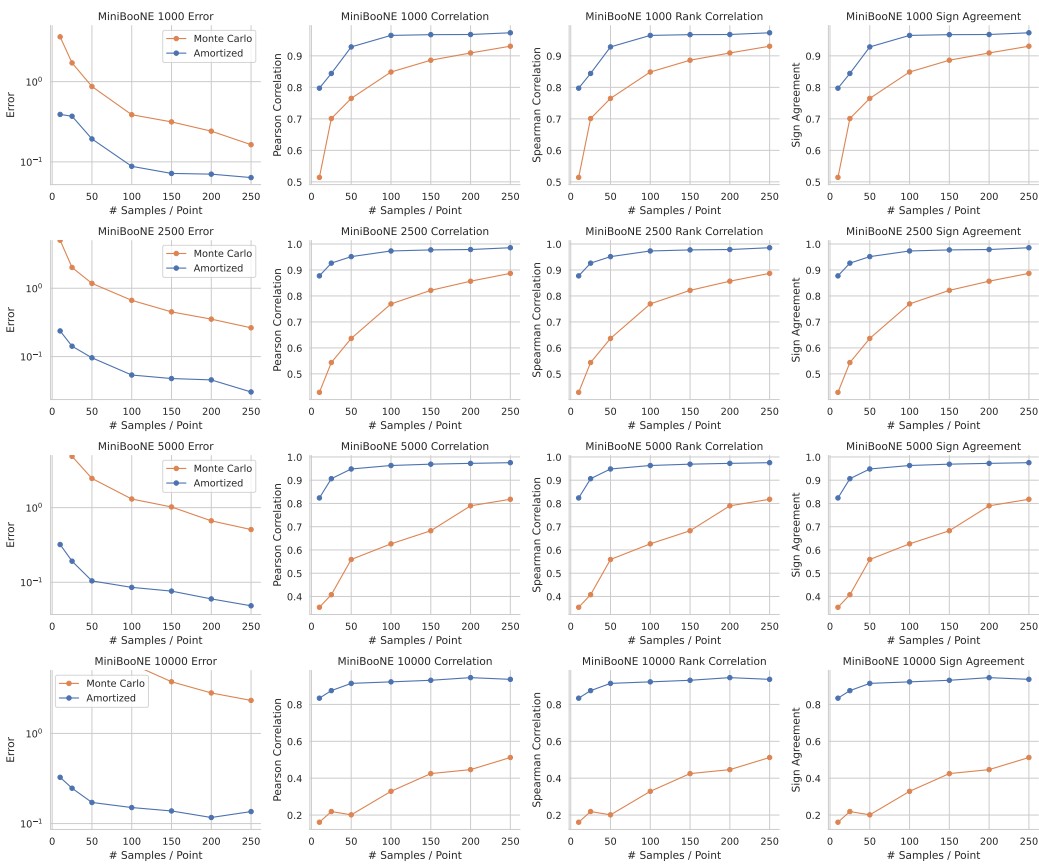

Figure 17: Amortized data valuation for the MiniBooNE dataset with different numbers of training data points (1K to 10K). We show four metrics: squared error (normalized so that the mean valuation scores has error equal to 1), Pearson correlation, Spearman correlation, and sign agreement.

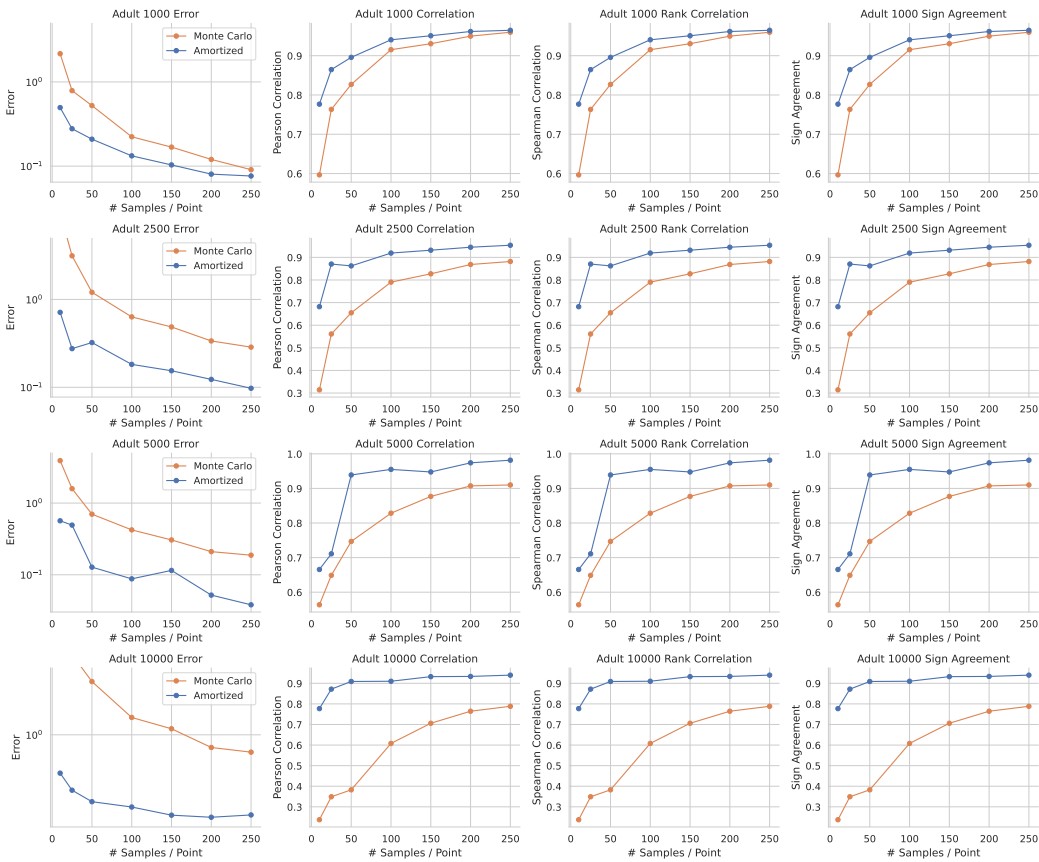

Figure 18: Amortized data valuation for the adult dataset with different numbers of training data points (1K to 10K). We show four metrics: squared error (normalized so that the mean valuation scores has error equal to 1), Pearson correlation, Spearman correlation, and sign agreement.

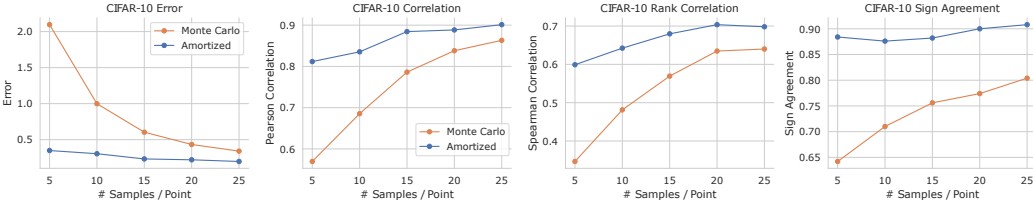

Figure 19: Distributional data valuation for CIFAR-10 with 50K data points. We generate Monte Carlo estimates and amortized estimates with different numbers of samples per data point, and the scores are compared to ground truth values using four metrics.

We also provide more detailed performance metrics for CIFAR-10: Figure 19 shows an expanded version of Figure 5, and we observe that amortization improves upon the Monte Carlo estimator across all four performance metrics.

Next, we consider the use of CIFAR-10 data valuation scores in two downstream tasks. First, we attempt to identify mislabeled examples, which we expect should have large negative valuation scores. Figure 20 shows how quickly each method identifies negative examples when we sort the scores from lowest to highest: our amortized estimates that train with just 5 samples provide the highest accuracy, outperforming Monte Carlo estimates that use as many as 100 samples. This result is consistent across different levels of label noise, where we randomly flip either 10%, 25% or 50% of the labels.

Table 2 performs a similar analysis involving mislabeled examples: we expect these examples to have lower scores than correctly labeled examples, so we use the estimated scores to calculate the AUROC, the AUPR, and the portion of mislabeled examples with negative scores. Across the different levels of label noise, our amortized estimates achieve the best performance according to all three metrics. The strong performance of our amortized data valuation scores is largely due to the improved estimation accuracy, but we expect that it is also due to the valuation network being trained on raw images: the noisy labels are derived from training logistic regression models on pretrained ResNet-50 embeddings, which can lead to somewhat arbitrary valuation score differences between semantically similar examples, and the valuation network may learn a more generalizable notion of data value.

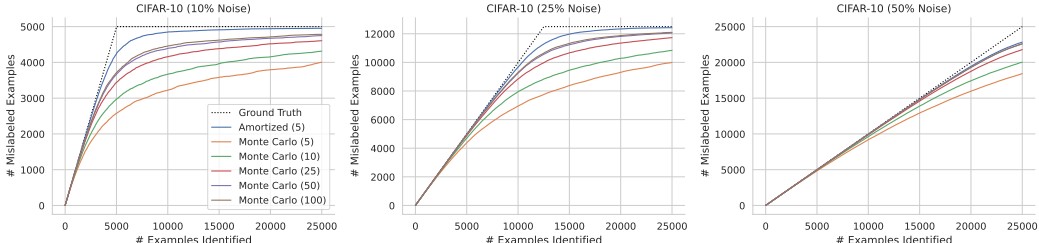

Figure 20: CIFAR-10 mislabeled example identification with different amounts of label noise. We compare the number of mislabeled examples among the lowest-scoring data points, where the amortized model uses just 5 samples for training and the Monte Carlo estimator uses between 5 and 100 samples.

Table 2: Mislabeled example identification accuracy for CIFAR-10 with different amounts of label noise.

|  | 10% Noise | | | 25% Noise | | | 50% Noise | | |
|---|---|---|---|---|---|---|---|---|---|
|  | AUROC | AUPRC | Negative | AUROC | AUPRC | Negative | AUROC | AUPRC | Negative |
| Amortized (5) | 0.981 | 0.917 | 0.974 | 0.985 | 0.964 | 0.973 | 0.972 | 0.973 | 0.915 |
| Monte Carlo (5) | 0.782 | 0.515 | 0.757 | 0.802 | 0.687 | 0.758 | 0.815 | 0.829 | 0.708 |
| Monte Carlo (10) | 0.844 | 0.619 | 0.826 | 0.867 | 0.784 | 0.821 | 0.883 | 0.892 | 0.769 |
| Monte Carlo (25) | 0.906 | 0.738 | 0.883 | 0.931 | 0.877 | 0.888 | 0.943 | 0.947 | 0.841 |
| Monte Carlo (50) | 0.934 | 0.794 | 0.909 | 0.956 | 0.918 | 0.918 | 0.965 | 0.967 | 0.878 |
| Monte Carlo (100) | 0.941 | 0.808 | 0.918 | 0.960 | 0.923 | 0.922 | 0.965 | 0.967 | 0.878 |

Finally, we experiment with using the data valuation scores to improve the dataset. Using the version with 25% label noise, we experiment with several possible modifications to the dataset, where in each case we train 5 models and report the mean cross entropy loss. First, we consider a perfect filtering of the data where we remove all mislabeled examples. Next, we remove different numbers of examples chosen uniformly at random. We also consider removing the examples with the lowest scores according to the Monte Carlo estimates with 10 samples. Finally, we test two possible approaches based on the amortized valuation model: (i) we filter out the lowest scoring examples ("Amortized filtering"), similar to how we use the noisy Monte Carlo estimates; (ii) we attempt to correct the lowest scoring examples using the class that the valuation model predicts to be most valuable ("Amortized cleaning"), which is a unique capability enabled by our valuation model that predicts valuation scores simultaneously for all possible classes (see Appendix F). The results of this experiment are shown in Figure 21. We see that removing samples at random hurts the model's performance relative to using all the data, that filtering with the amortized estimates outperforms filtering with the Monte Carlo estimates, and that cleaning the estimates is the best approach by a narrow margin.

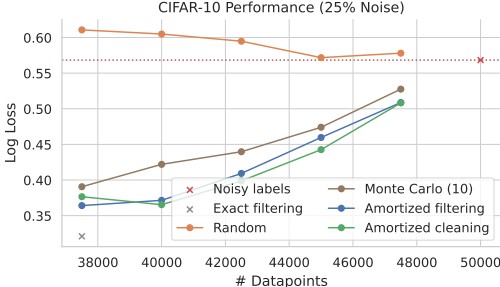

Figure 21: CIFAR-10 performance when training data is removed or adjusted based on estimated valuation scores.

## H   Broader Impact

Our proposed method aims to make many computationally challenging XML tasks feasible and scalable to large datasets. We expect our research to contribute to a better understanding of ML models, enhancing their transparency and explainability. Additionally, our research could help understand model fairness properties if our method is used to identify undesirable or unfair dependencies of ML models on input features related to protected attributes like race or gender. The potential downside is that that if users overly trust our method and rely solely on it for these tasks without careful usage, it could lead to misunderstanding of models and result in harmful outcomes.

