# OpenReview forum: "Stochastic Amortization: A Unified Approach to Accelerate Feature and Data Attribution"
_NeurIPS.cc/2024/Conference — NeurIPS 2024 poster_

### Official Review · Reviewer_hGpm · 2024-06-17

**Soundness:** 3
**Presentation:** 3
**Contribution:** 3
**Rating:** 6
**Confidence:** 4

**Summary:**

The paper proposes using stochastic amortization to speed up feature attribution and data attribution. This is applicable when the attribution technique is expensive to compute exactly (e.g. LIME), and when unbiased estimators of the attributions exist. Specifically, the paper proposes to, for a given machine learning model (e.g. Resnet) and a dataset of inputs (e.g. cifar) on which we want to compute attribution (e.g. Shapley feature values), obtain noisy estimates of the attributions (e.g. by sampling permutations), and then train a least square regression model to predict these noisy estimates from the input. The paper evaluates this method against the ground-truth attribution values for Shapley values on Imagenette,  data valuation shapley on adult census and MiniBooNE datasets, and Distributional data valuation on the CIFAR-10 dataset.

**Strengths:**

- The paper clearly explains its motivations and approach
- Using the amortized model improves performance over the initial noisy labels used for training, clearly showing the benefits of using an amortized method
- The paper provides some theoretical arguments for using unbiased estimators.

**Weaknesses:**

- The related work section could benefit with a more direct and concrete comparison with prior work. For example, the paper says that "while there are works that accelerate data attribution with datamodels, we are not aware of any that use amortization". It is not clear to me in what "amortization" mean here exactly --- in what way does your work perform amortization that datamodels do not?
- In addition, the experimental results lack baselines that compare with prior amortization work. The related work section describes a number of prior works (citations 50, 86, 18, 14) that all seem to be about computing Shapley values with amortization.
- Figures are missing error bars

**Questions:**

- S5.1: would be interested to look at fig 3 right with a log scale on the y axis. Amortization seems to benefit with increasing dataset size, and I wonder what the scaling looks like
- Related to that, a key hyperparameter in your amortization set up is to choose between spending compute to get fewer but more accurate labels vs more noisy labels. Is there a way of determining an optimal trade-off?
- How does the paper's method compare with prior works? This is a crux for me.

**Limitations:**

The authors adequately discussed technical limitations of their amortization approach in the appendix.

---

> ### Author Rebuttal · Authors · 2024-08-07
>
> Thank you very much for your review, we've responded to your points below.
>
> > The related work section could benefit with a more direct and concrete comparison with prior work. For example, the paper says that "while there are works that accelerate data attribution with datamodels, we are not aware of any that use amortization". It is not clear to me in what "amortization" mean here exactly --- in what way does your work perform amortization that datamodels do not?
>
> We can clarify this in the paper, but datamodels isn’t a form of amortization because it doesn’t learn a neural network that takes two data points as input and predicts the attribution score. It instead learns a linear model that takes a one-hot representation of the training dataset as the input, and predicts the loss for an inference example. The attribution scores are the learned coefficients of that model, not its predictions. Crucially, you can’t use the datamodels linear model to estimate attributions for new data points. Appendix B contains a brief discussion on how amortization would be implemented for datamodels, including a neural network $\zeta(z, x; \theta)$ whose output is the estimated attribution score.
>
> > In addition, the experimental results lack baselines that compare with prior amortization work. The related work section describes a number of prior works (citations 50, 86, 18, 14) that all seem to be about computing Shapley values with amortization. [...] How does the paper's method compare with prior works? This is a crux for me
>
> Thanks for this request, you raise a good point. There are fewer relevant comparisons than it might seem, but you’re correct that we missed the comparison with FastSHAP. We’ve now fixed this, as we describe below, and we can also clarify why certain other papers aren’t useful comparisons.
>
> First, we’ll address [86] and [14]. Setting aside some minor details, these are basically versions of stochastic amortization with much less noisy labels. They train explainer models to predict SVs using a supervised objective and targets computed with relatively high accuracy: [86] uses 10$\times$ the number of samples we do, and [14] claims to use either exact labels or orders of magnitude more than us depending on the experiment. (Besides these details, [86] also includes an unnecessary normalization step and regularization term, and [14] suggests a custom pre-training approach, which is an orthogonal contribution.) Given that these are basically much less noisy versions of stochastic amortization, a direct comparison wouldn’t be very interesting: using more accurate labels will yield a more accurate explainer, but at the cost of much more computation. This type of comparison is already shown in our work (e.g., Figure 3 left), only we don’t use as many samples as those works even in our most compute-intensive settings. Our goal here was to explore the other direction and demonstrate the ability to use noisier supervision.
>
> Next, [50, 18] both explore SV amortization using a custom weighted least squares objective, which we’ll refer to as “FastSHAP.” The main differences from our approach are that FastSHAP uses new samples for every gradient step and requires a complex training procedure, whereas we use pre-computed noisy labels from any unbiased estimator and train with MSE loss. Notably, the FastSHAP explainer model *can’t make predictions without access to the original model’s predictions* due to an output transformation required by its objective (“additive efficient normalization” [18]). Our approach is simpler to train, more flexible in the supervision, and the explainer can operate as a standalone model.
>
> Still, it’s worth including the comparison. To provide a compute-matched comparison, we implemented FastSHAP and measured the error from each approach as a function of the total FLOPs (including the cost of querying the original model, plus the cost of training the amortized model), using 32 subset samples/step for FastSHAP as in [18]. We found that the error and correlation with the ground truth is very similar between FastSHAP and our approach, with both significantly more accurate than KernelSHAP, but that FastSHAP is marginally more accurate ([plot link](https://imgur.com/a/5oGOjTK)). The result doesn’t seem conclusive, however, because our approach can be implemented with any noisy oracle, including more advanced ones that we didn’t test here; future work testing other unbiased estimators could find that stochastic amortization is more accurate. Overall, we believe our work is still valuable in presenting a simpler/more flexible alternative that reaches basically the same accuracy, and which can be easily applied to other XML methods like data valuation.
>
> > Figures are missing error bars
>
> Thanks for noticing this. We didn’t perform multiple trials due to the high computational cost, specifically the time required to run the Monte Carlo estimators for a small-to-moderate number of samples for all training data points. Given the wide margin of improvement we saw in the experiments (often an order of magnitude lower error), we didn’t think it would add much value to repeat the experiments multiple times. But we’ll try to fix this and prepare error bars in time for the final version of the paper.
>
> > S5.1: would be interested to look at fig 3 right with a log scale on the y axis. Amortization seems to benefit with increasing dataset size, and I wonder what the scaling looks like
>
> That’s a good idea, we made a version of the plot with both axes shown in log-scale ([plot link](https://imgur.com/a/lWC1YPR)). We hoped to see a log-linear trendline, because this is similar to a Kaplan et al. (2020)-style scaling curve, and it’s close but not quite linear. The error for the smallest dataset size is a bit too high, perhaps because it’s the least reliable point. If it’s helpful, we can change the plot in the final version to use log-scale on both axes.

---

> > ### Author Response · Authors · 2024-08-07
> > **Rebuttal by Authors (cont.)**
> >
> > > Related to that, a key hyperparameter in your amortization set up is to choose between spending compute to get fewer but more accurate labels vs more noisy labels. Is there a way of determining an optimal trade-off?
> >
> > That’s a good question, and one that we didn’t try to answer conclusively here. Our goal was to show that training with noisy labels works, that it tolerates surprisingly high noise levels, and that it’s applicable to many XML tasks (including both feature attribution and data valuation). Investigating the quality-quantity tradeoff in noisy labels is an interesting question and a natural subject for future work, but addressing it thoroughly would be a bit too much work for our current paper. We are happy to mention this in the conclusion and provide some preliminary thoughts on how one should approach it.
> >
> > Intuitively, the optimal point can’t be at either extreme: a very small number of exact labels would be unlearnable, as would a large number of extremely noisy labels. Finding the optimal point requires reasoning about how well neural networks learn as a function of the dataset size and label noise, which is dataset- and model-dependent and likely best to approach empirically. One idea is to 1) measure the estimation error as a function of dataset size and number of Monte Carlo samples (i.e., run stochastic amortization under a range of settings), 2) fit the observed errors using a simple analytic function (in the style of recent scaling laws), and 3) use this to estimate the optimal trade-off (sort of like Chinchilla’s compute-optimal training). This approach seems reasonable and straightforward, but it would require many training runs and is beyond the scope of our current paper.

---

> ### Comment · Reviewer_hGpm · 2024-08-09
>
> Thank you for the detailed response. Just an additional clarification about the comparison with prior work - would you agree that the technique you use is a simplification of previous techniques, which is enabled by the observation that stochastic amortization can tolerate more noise than previous methods assumed?

---

> > ### Author Response · Authors · 2024-08-10
> > **Author response**
> >
> > Thanks for your response, we hope we resolved the concerns mentioned in your review. As for whether our technique is a simplification of prior works – you could describe it that way, that's basically correct regarding [86, 14]: the fact that amortization tolerates high noise levels enables our more practical approach, although the mathematical justification makes it perhaps less conceptually simple than training with exact targets. As for FastSHAP, our approach is simpler, but it’s not exactly a simplification of [50, 18] in the sense that the objective is derived differently. (FastSHAP is based on a weighted least squares view of SVs, whereas our approach is based on SVs being the expectation of a Monte Carlo estimator.)

---

> > > ### Comment · Reviewer_hGpm · 2024-08-10
> > >
> > > Thank you for the clarification. I encourage the authors to make that connection with prior work clear in the paper (i.e. that the key observation is that stochastic amortization can tolerate more noise than previous methods assumed). I believe, already, a lot of the empirical tests in the current is aimed at studying how much noise can be tolerated.
> > >
> > > Because you have committed to a more direct comparison with prior work, and have shown empirical tests of prior baselines, I will raise my score accordingly.

---

### Official Review · Reviewer_cihK · 2024-07-11

**Soundness:** 3
**Presentation:** 2
**Contribution:** 3
**Rating:** 5
**Confidence:** 3

**Summary:**

This paper presents a fast prediction explanation approach for machine unlearning models that approximates traditional explanation approaches by using an neural network. The network is trained with noisy labels (so called stochastic amortization) such that it learns to approximate the prediction explanations of various approaches, including Shaply, LIME, and others. Experiments covers multiple explanation approaches from feature explanation to data instance explanation, showing effectiveness of the approximation.

**Strengths:**

Using network to approximate prediction explanation of various explanation methods seem an interesting idea.
Using noisy labels to train the approximation model is also well justified with corresponding proofs.
Experiments covers the various scenarios from feature to instance based explanation approaches.

**Weaknesses:**

It is concerning to use neural networks to explaining the predictive behaviour of another machine learning model. In fact, how do we know whether the explanation itself is trustworthy? If the prediction explanation is given to an stakeholders, is it possible to distinguish the potential concern between target machine learning model's trustworthiness and the approximative explainer? This concern needs to be addressed further to let the proposed approach practically useful.

Partial experimental results show the proposed model may not be a good approximation approach; the results can be interpreted as the approximation fails to provide correct explanation to feature importance (Section 5.1). It causes me wonder if there is a practical value for the proposed approach if it is fast but not accurate.

**Questions:**

How do you prove the proposed approximation is, in general, faithful to the ground-truth explanation results from the actual explanation models?
Why would the proposed approach to be useful if the explanation is off from the ground-truth?

**Limitations:**

I don't see limitations. But, as the proposed approach is about XAI, it should be sound in terms of improving the trustworthiness of prediction instead of introducing another layer of uncertainty.

---

> ### Author Rebuttal · Authors · 2024-08-07
>
> Thank you very much for your review, we've responded to your points below.
>
> > It is concerning to use neural networks to explaining the predictive behaviour of another machine learning model. In fact, how do we know whether the explanation itself is trustworthy?
>
> Thanks for raising this question—by “trustworthy”, we assume you’re asking how we know that the amortized estimates are accurate. Ensuring high accuracy is very important, and our theory and experiments are all about showing that the error from amortization is small despite training with noisy labels. You can see that across all our experiments, amortization is always more accurate than Monte Carlo estimation for an equal computational budget, in most cases by a wide margin. For example, Figure 3 shows that the error from amortization is 1-2 orders of magnitude smaller than the standard KernelSHAP approach. This high accuracy (low error) is what makes the explanations trustworthy.
>
> We aren’t sure if this is what you meant, but you may be getting at the fact that our analysis focuses on average error across data points rather than worst-case error. Minimizing average-case error is useful in many scenarios, e.g., identifying mislabeled data points with data valuation scores, understanding a classifier by visually checking many feature attributions, and identifying common shortcuts/confounders. There’s an established literature on this type of approach, but there may be cases where Monte Carlo estimation with a large computational budget is preferable to provide guarantees for each data point's explanation.
>
> > Partial experimental results show the proposed model may not be a good approximation approach; the results can be interpreted as the approximation fails to provide correct explanation to feature importance (Section 5.1).
>
> We aren’t sure what you mean and which of these results reflect that the approximation is inaccurate. The examples in Figure 2 are qualitatively quite accurate, and the metrics in Figure 3 show that amortization achieves significantly lower error than the non-amortized estimates. Note that some of the metrics are shown in log-scale, which may make the error appear far from zero when it’s actually quite small. If you’re expecting to see zero error, note that these XML methods are never used with zero error due to the high computational cost – the only option is to use approximations, that’s why there’s so much work on efficient approximation algorithms.
>
> > How do you prove the proposed approximation is, in general, faithful to the ground-truth explanation results from the actual explanation models? Why would the proposed approach to be useful if the explanation is off from the ground-truth?
>
> Thanks for bringing this up. We establish the faithfulness of the approximation empirically: the estimation error is often small in our experiments, where we substitute the ground truth for estimates obtained using a large number of samples (e.g., 1M samples for KernelSHAP, see details in Appendix F). The error from amortization is always smaller than the Monte Carlo error given an equal computational budget, so our approach seems strictly preferable.
>
> The proposed approach wouldn’t be useful if the explanation was significantly off from the ground truth. However, a small degree of error is generally tolerable in practice, as evidenced by the fact that the community relies entirely on approximation algorithms, which are rarely run long enough to reach exact convergence (see for example the number of samples in KernelSHAP https://github.com/shap/shap/blob/master/shap/explainers/_kernel.py#L192). Our focus on fast/accurate approximation is not unusual, it’s an established line of work in the literature.

---

> > ### Comment · Reviewer_cihK · 2024-08-11
> >
> > Thanks for your answers. While my concerns of faithfulness of the NN approximation are not fully address, I think the paper may add value to the literature. Hence, I bumped my rating.

---

### Official Review · Reviewer_x1Ww · 2024-07-12

**Soundness:** 3
**Presentation:** 4
**Contribution:** 3
**Rating:** 7
**Confidence:** 3

**Summary:**

In this paper, the authors introduce a framework termed stochastic amortization that can accelerate computationally expensive explainable machine learning (XML) tasks by training models with noisy but unbiased labels. They provide theoretical analysis that shows that unbiased noisy labels allow learning the correct function. Empirically, the authors demonstrate the effectiveness of this approach on several XML tasks, including Shapley value feature attributions, Banzhaf values, LIME, and data valuation, achieving speedups over traditional per-example computation and improvements over simply using noisy labels.

**Strengths:**

- The paper is well-written, with clear motivations and a thorough discussion of related work. The technical details appear sound and well-presented.
- The proposed approach offers a simple yet effective framework for accelerating computationally expensive XML methods, potentially enabling their application to large-scale datasets.
- The authors provide code (with good documentation) and detailed experimental information in the Appendix for reproducability.
- Limitations of the proposed approach are well discussed, demonstrating a balanced perspective.
- Comprehensive experiments across multiple XML tasks (Shapley values, Banzhaf values, LIME, and data valuation) consistently demonstrate the benefits of stochastic amortization over per-example computation.

**Weaknesses:**

While the concept of amortization for Shapley value prediction is not entirely new, this paper makes a good contribution through its use of noisy labels and the accompanying theoretical analysis. I don't find any major weaknesses that would significantly detract from the paper's value, and I recommend acceptance. Reflecting my limited familiarity with some aspects of the relevant literature, I set my confidence to 3.

**Questions:**

- For applying stochastic amortization to a new XML task, how should practitioners determine an appropriate error level from a practical perspective?
- Regarding line 99, "while there are works that accelerate data attribution with datamodels, we are not aware of any that use amortization". Could datamodels themselves be considered a form of amortized optimization (linear regression model)?
- What guidelines can you provide for defining the architecture for amortization (e.g., ResNet) in practice? Did you observe significant differences with various architectures? Should the architecture be similar to the base model?
- In Figure 4 (right), why does the correlation decrease for MC as the number of data points increases (intuitively, I expected correlation to remain relatively flat for both MC and amortized approaches, assuming a constant number of samples per point)?

**Limitations:**

The authors adequately addressed the limitations of the proposed approach.

---

> ### Author Rebuttal · Authors · 2024-08-07
>
> Thank you very much for your review, we've responded to your points below.
>
> > For applying stochastic amortization to a new XML task, how should practitioners determine an appropriate error level from a practical perspective?
>
> Thanks for asking this question. Our recommendation is to create a small validation set with near-exact estimates, for example by running a standard Monte Carlo estimate for many iterations, and then use this to determine what level of label noise achieves low enough error. This is how we evaluated estimation accuracy in our experiments. As for how to define “low enough error”, this depends on the task and its error tolerance, e.g., feature attribution can tolerate some error if its goal is to visually show humans the important part of an image.
>
> > Regarding line 99, "while there are works that accelerate data attribution with datamodels, we are not aware of any that use amortization". Could datamodels themselves be considered a form of amortized optimization (linear regression model)?
>
> That’s a good question and one that we get a lot. Datamodels isn’t a form of amortization because it doesn’t learn a neural network that takes two data points as input and predicts the attribution score. It instead learns a linear model that takes a one-hot representation of the training dataset as the input, and predicts the loss for an inference example. The attribution scores are the learned coefficients of that model, not its predictions. Crucially, you can’t use the datamodels linear model to estimate attributions for new data points. Appendix B contains a brief discussion on how amortization would be implemented for datamodels, including a neural network $\zeta(z, x; \theta)$ whose output is the estimated attribution score.
>
> > What guidelines can you provide for defining the architecture for amortization (e.g., ResNet) in practice? Did you observe significant differences with various architectures? Should the architecture be similar to the base model?
>
> That’s an interesting question. The best architecture for each problem is likely a function of how much expressive power you need and how much training data is available, similar to any setting where you use deep learning. Using the pre-trained base model is natural because it already extracts many relevant features, which can help with efficient fine-tuning, but it’s possible that in some cases you could benefit from a larger architecture or that you could get sufficient accuracy with a smaller one. In our experiments, we only tried training the base model architecture.
>
> > In Figure 4 (right), why does the correlation decrease for MC as the number of data points increases (intuitively, I expected correlation to remain relatively flat for both MC and amortized approaches, assuming a constant number of samples per point)?
>
> This was somewhat surprising to us as well. We believe it’s because as you increase the size of the dataset, the variance of each data point’s marginal contributions becomes larger relative to the expectation (reflecting a lower signal-to-noise ratio), and this leads to decreasing correlation even with a fixed number of samples. This observation is related to, but not exactly the same as one in the Beta Shapley paper (see Figure 1 in https://arxiv.org/abs/2110.14049).

---

> > ### Comment · Reviewer_x1Ww · 2024-08-08
> > **Response**
> >
> > Thank you for the comments. I read the author's response and other reviewers' comments. The clarification on the datamodels was helpful, and it would be great if you could add this explicit answer somewhere (I have read Appendix B but was still unsure). Overall, I am satisfied that the authors addressed all my questions.

---

> > > ### Author Response · Authors · 2024-08-10
> > > **Author response**
> > >
> > > Thanks very much for reading our rebuttal and for your response. We'll make sure to add that clarification about datamodels to the paper, especially since hGpm had the same question.

---

### Official Review · Reviewer_GyYk · 2024-07-18

**Soundness:** 3
**Presentation:** 3
**Contribution:** 3
**Rating:** 6
**Confidence:** 5

**Summary:**

This paper proposes a stochastic amortization framework for efficiently estimating feature attribution and data attribution values. The idea is to learn a parameterized model from noisy while unbiased samples of the value to be estimated. In comparison to naive Monte Carlo sampling, this amortized estimation is empirically more efficient when achieving similar estimation accuracy. The authors conducted experiments on a diverse set of feature attribution and data attribution problems.

**Strengths:**

- This paper tackles the computational efficiency issue in a set of feature attribution and data attribution problems, which is an important problem that could have significant impact to the area of explainable machine learning (XML).
- This paper abstracts the paradigm of learning a parameterized function to estimate the attribution values in some XML methods with a more general amortized estimation framework, making it possible to extend this approach to more XML methods that conventionally rely on naive Monte Carlo estimators.
- The authors conducted comprehensive experiments in diverse settings to demonstrate the effectiveness of the stochastic amortization.

**Weaknesses:**

- The theoretical analysis, especially Theorem 1, is not directly relevant to the main argument of this paper, i.e., stochastic amortization is more efficient than naive Monte Carlo. IMO this Theorem 1 is a bit distracting than being helpful for this paper. Furthermore, the theoretical result about the unbiasedness in lines 128 - 132 only concerns the expected loss, $\tilde{\mathcal{L}}_{reg}$, which doesn't directly imply the property of the empirically learned parameterized model. Overall, the advantage of stochastic amortization over Monte Carlo estimation is established empirically instead of theoretically, while the current presentation reads a bit misleading on this point.
- In most of the experiments, only relatively naive Monte Carlo estimators compared as baselines. However, for certain specific problems, more efficient estimators may be available. For example, in the context of Data Shapley, how does the stochastic amortization compare with Gradient Shapley proposed in Ghorbani and Zou [33], or the compressive-sensing-based method proposed in Jia et al. (2019).

References
- Jia et al. (2019) Towards Efficient Data Valuation Based on the Shapley Value. AISTATS 2019.

**Questions:**

See Weaknesses.

**Limitations:**

None.

---

> ### Author Rebuttal · Authors · 2024-08-07
>
> Thank you very much for your review, we've responded to your points below.
>
> > The theoretical analysis, especially Theorem 1, is not directly relevant to the main argument of this paper, i.e., stochastic amortization is more efficient than naive Monte Carlo.
>
> Thanks for bringing up this concern. To clarify, Theorem 1 shows that when training the amortized model $a(b; \theta)$ with SGD and noisy labels $\tilde a(b)$, the convergence rate is affected by label noise via $\text{N}_q(\tilde a)$. This means that noisier labels make training slower, which is an important theoretical point to understand about stochastic amortization: label noise is the key difference between stochastic amortization and the noiseless version, so it seems important to include Theorem 1 and highlight its role in slowing optimization.
>
> However, you’re correct that Theorem 1 doesn’t consider the stochastic amortization vs. Monte Carlo comparison. We found that harder to characterize in a satisfactory way and therefore relied on empirical results. But just to show that it’s possible, here’s a result that follows from Theorem 1 and demonstrates the advantage of sharing information across data points (as compared to Monte Carlo estimation, which treats each point independently):
>
> When we take $T$ SGD steps to update the amortized model, Theorem 1 shows that the estimation error shrinks at a rate of $\mathcal{O}(1/T)$. If we instead used the exact same training labels $\tilde a(b)$ as one-sample Monte Carlo estimates for each training data point $b$, our expected error across those would be $\text{N}_p(\tilde a)$ (this is a quick result to show). Crucially, this error does not shrink with $T$, as we just obtain (bad) estimates for more data points. So, we can see that for a sufficient number of gradient steps $T$, which depends on the label noise and data distribution, our error from the amortized model will eventually become lower than the error from the single-sample Monte Carlo estimates. And notably, the model can be used to generate estimates for any data point $b$ rather than only the fixed set of training data points. Hopefully this helps clarify the advantage of amortization over per-example Monte Carlo estimation. We’re happy to add the result to the paper, but we think the empirical results are most convincing for this point.
>
> > Furthermore, the theoretical result about the unbiasedness in lines 128 - 132 only concerns the expected loss, $\tilde{\mathcal{L}}_{\text{reg}}(\theta)$, which doesn't directly imply the property of the empirically learned parameterized model.
>
> Thanks for pointing this out. Your concern seems to be that our theoretical results don’t address the generalization gap between the empirical and population version of $\tilde{\mathcal{L}}_{\text{reg}}(\theta)$. That’s correct, but if you’re concerned about a large generalization gap in practice, generalization is famously not much of an issue with deep learning (https://arxiv.org/abs/1611.03530). We use standard approaches like a validation set to avoid overfitting, and some existing theoretical results also apply here (https://arxiv.org/abs/1901.08584). We can mention these points in the paper, but extending our theory to include the generalization gap doesn’t seem helpful given the technical complexity of the topic.
>
> > In most of the experiments, only relatively naive Monte Carlo estimators compared as baselines. [...] For example, in the context of Data Shapley, how does the stochastic amortization compare with Gradient Shapley proposed in Ghorbani and Zou [33], or the compressive-sensing-based method proposed in Jia et al. (2019).
>
> Thanks for raising this point. We prioritized breadth of experiments across XML techniques rather than depth in a single one, and as a result we can’t explore the full range of Monte Carlo estimators. There are numerous choices for both data valuation and feature attribution, several of which are discussed in Section 4 and Appendix E; since our theory applies to any unbiased estimator, we can’t see a reason why other options wouldn’t work. However, the two techniques you mentioned are different because they’re both *biased estimators* (Gradient Shapley because it uses a cheap proxy for the marginal contribution, and Compressive Permutation because it relies on sparsity assumptions). Our work is focused primarily on unbiased estimators, so adding these doesn’t seem like a priority, but it could be an interesting topic for other work to explore. One immediate intuition for such work is that with these approaches, stochastic amortization would learn to predict the expectation of those biased estimators, which may or may not be close to the true Data Shapley values.

---

> ### Comment · Reviewer_GyYk · 2024-08-08
> **Thanks for the response**
>
> I appreciate the authors' response. I would encourage the authors to at least add a remark after Theorem 1 clarifying the gap between Theorem 1 and the main claims of the advantage of the proposed method (which are evaluated empirically). This will help avoid potential confusions by readers like myself.
>
> PS: Part of the confusion comes from the paper abstract: "Through theoretical analysis of the label noise and experiments with various models and datasets, we show that this approach significantly accelerates several feature attribution and data valuation methods", which sets up an expectation that Theorem 1 is directly about the advantage of the proposed method.
>
> Overall I think this is a solid paper worth publication at NeurIPS. And this assessment has already been reflected by my previous rating.

---

> > ### Author Response · Authors · 2024-08-10
> > **Author response**
> >
> > Thanks very much for reading our rebuttal and for your response. We understand the confusion about Theorem 1, thanks for pointing out that sentence in the abstract. We'll make sure to clarify that and add a remark after Theorem 1 to avoid confusion for other readers.

---

### Author Rebuttal · Authors · 2024-08-07

Thank you to all reviewers for your detailed feedback. We have addressed your points in individual responses below. If you find that we have resolved your concerns, we would greatly appreciate it if you would consider revising your score.

---

### Decision · Program_Chairs · 2024-09-25

**Decision:**

Accept (poster)

**Comment:**

The reviewers were generally positive, and the authors did a good job addressing their concerns during the response period. Please make sure that the changes you promised to the reviewers are incorporated into the current version of the paper.